# Multiplexed RNA profiling by regenerative catalysis enables blood-based subtyping of brain tumors

Yan Zhang[1,2,12], Chi Yan Wong [1,3,12], Carine Z. J. Lim[1,2,12], Qingchang Chen[1,2], Zhonglang Yu [1,2], Auginia Natalia [1,2], Zhigang Wang[1], Qing You Pang[4], See Wee Lim[4], Tze Ping Loh[1,5], Beng Ti Ang[6,7], Carol Tang [4,7,8] & Huilin Shao [1,2,9,10,11] ✉

Current technologies to subtype glioblastoma (GBM), the most lethal brain tumor, require highly invasive brain biopsies. Here, we develop a dedicated analytical platform to achieve direct and multiplexed profiling of circulating RNAs in extracellular vesicles for blood-based GBM characterization. The technology, termed 'enzyme ZIF-8 complexes for regenerative and catalytic digital detection of RNA' (EZ-READ), leverages an RNA-responsive transducer to regeneratively convert and catalytically enhance signals from rare RNA targets. Each transducer comprises hybrid complexes – protein enzymes encapsulated within metal organic frameworks – to configure strong catalytic activity and robust protection. Upon target RNA hybridization, the transducer activates directly to liberate catalytic complexes, in a target-recyclable manner; when partitioned within a microfluidic device, these complexes can individually catalyze strong chemifluorescence reactions for digital RNA quantification. The EZ-READ platform thus enables programmable and reliable RNA detection, across different-sized RNA subtypes (miRNA and mRNA), directly in sample lysates. When clinically evaluated, the EZ-READ platform established composite signatures for accurate blood-based GBM diagnosis and subtyping.

Identification of disease subtypes has broad applications in patient stratification and treatment selection[1,2]. For instance, primary adult glioblastoma (GBM), the most common type of malignant brain tumor, is morphologically identical; however, recent consortium efforts have demonstrated extensive molecular heterogeneity in the tumor and highlighted its potential role in mediating inter-patient differences to treatment[3,4]. In particular, GBM can be transcriptionally characterized by three glioma-intrinsic (GI) subtypes, namely proneural, classical and mesenchymal; these subtypes have been shown to associate with differential prognosis and treatment response[5,6]. Most notably, patients

[1]Institute for Health Innovation & Technology, National University of Singapore, Singapore, Singapore. [2]Department of Biomedical Engineering, College of Design and Engineering, National University of Singapore, Singapore, Singapore. [3]Department of Medicine, Yong Loo Lin School of Medicine, National University of Singapore, Singapore, Singapore. [4]Neuro-Oncology Research Laboratory, Department of Research, National Neuroscience Institute, Singapore, Singapore. [5]Department of Laboratory Medicine, National University Hospital, Singapore, Singapore. [6]Department of Neurosurgery, National Neuroscience Institute, Singapore, Singapore. [7]Duke-National University of Singapore Medical School, Singapore, Singapore. [8]School of Biological Sciences, Nanyang Technological University Singapore, Singapore, Singapore. [9]National Neuroscience Institute, Singapore, Singapore. [10]Department of Surgery, Yong Loo Lin School of Medicine, National University of Singapore, Singapore, Singapore. [11]Institute of Molecular and Cell Biology, Agency for Science, Technology and Research, Singapore, Singapore. [12]These authors contributed equally: Yan Zhang, Chi Yan Wong, Carine Z.J. Lim. ✉e-mail: huilin.shao@nus.edu.sg

who demonstrated subtype switching from non-mesenchymal (i.e., proneural and classical) to mesenchymal profile showed poorer survival compared to patients without the mesenchymal subtype switching[7]. Given the dynamic nature of GBM subtypes as well as their differential prognosis, minimally-invasive approaches that can characterize and monitor disease subtypes could bring forth clinical opportunities for personalized treatment[8,9]. Nevertheless, current GBM subtyping requires invasive brain biopsies for RNA analysis; for serial monitoring, the complexities and morbidity of repeat biopsies render the approach clinically unacceptable.

Extracellular vesicles (EVs) have recently emerged as an attractive circulating biomarker for GBM tumors[10–12]. EVs are nanoscale membrane vesicles actively secreted by a variety of cells, especially by rapidly dividing cancer cells[13–15]. A promising blood biomarker, they readily cross the blood-brain barrier and carry diverse nucleic acids (e.g., short miRNA and long mRNA) from their parent cells. Despite this rich RNA composition for blood-based GBM subtyping, their clinical translation has been challenging, primarily due to limitations of existing technologies in measuring low concentration and large diversity of these circulating RNA targets[16,17]. For example, to detect scarce RNA biomarkers, conventional analysis (e.g., polymerase chain reaction, PCR) relies primarily on target amplification, where target nucleic acid sequences need to be extensively replicated before detection[18–20]. The approach requires careful sequence design (e.g., for short miRNA elongation and amplification) and extensive sample processing (e.g., reverse transcription and thermal cycling-based sequence amplification), thereby posing challenges for reliable measurement and multiplexed detection, especially for diverse RNA targets of different lengths and sequences. To improve the assay adaptability, several hybridization-based approaches[21,22] have been developed to reduce the dependence on nucleic acid amplification. Nevertheless, they commonly suffer from limited sensitivity, and cannot be easily applied for the analysis of rare RNA targets.

Addressing these challenges, here we develop a dedicated analytical technology to directly transduce RNA targets and catalytically enhance signaling responses, thereby enabling reliable quantitation of diverse circulating RNAs for GBM characterization. Named **e**nzyme **Z**IF-8 complexes for **re**generative and c**a**talytic **d**igital detection of RNA (EZ-READ), the technology leverages an RNA-responsive transducer to achieve direct activation and catalytic digital quantification. Each transducer comprises hybrid nanoparticle complexes – protein enzymes encapsulated within metal organic frameworks – to configure strong catalytic activity and robust protection, even against potential inhibition in complex biological environments. The transducer recognizes target RNA and activates regeneratively; upon RNA hybridization with the transducer, catalytic complexes are liberated, in a target-recyclable manner, leading to enhanced signal transduction. The released complexes can singly catalyze strong chemifluorescence reaction; when individually partitioned and reacted in a fractal branching microfluidic chip, they enable digital quantification of different RNA targets.

Harnessing its regenerative transduction and catalytic detection, the EZ-READ platform obviates the need for target amplification. It enables programmable and reliable RNA detection, across different-sized RNA subtypes (miRNA and mRNA), directly in minimally processed sample lysates. The technology establishes a sensitive limit of detection (<10 RNA copies) and can be completed within 30 min. Employing the EZ-READ platform, we measure miRNA and mRNA targets in blood samples of GBM patients and control subjects. Using these clinical measurements, we construct a multilayer decision model to establish circulating RNA signatures for GBM diagnosis and subtyping. The technology not only accurately diagnoses patients but also effectively distinguishes different tumor subtypes.

## Results
### EZ-READ platform
The EZ-READ platform leverages an RNA-responsive transducer to achieve two functional steps: regenerative signal transduction and catalytic digital quantification (Fig. 1a). The transducer comprises hybrid nanoparticle complexes covalently linked to magnetic beads via specific DNA probes (Fig. 1b). Within each nanoparticle complex, the enzyme horseradish peroxidase (HRP) is encapsulated in a metal organic framework (ZIF-8); this encapsulation (HRP@ZIF-8) not only configures the enzyme to achieve higher intrinsic catalytic activity, but also shields the enzyme against potential inhibition in complex biological environments (Supplementary Fig. 1a, b). The EZ-READ platform employs this potent catalytic transducer to achieve direct and sensitive RNA quantification. During regenerative signal transduction, RNA targets bind and trigger an enhanced release of HRP@ZIF-8 nanoparticles, in a target-recyclable manner. Specifically, upon RNA hybridization with the transducer (i.e., via the single-stranded DNA probe), duplex-specific nuclease (DSN)[23] recognizes the RNA-DNA duplex and cleaves only the DNA linker; this not only liberates the attached HRP@ZIF-8 nanoparticle, but also regenerates the RNA target, priming it for the next reaction (Supplementary Fig. 1c, d). To enable sensitive measurement of target transduction, exploiting the strong catalytic activity of the HRP@ZIF-8 nanoparticles, we develop a fractal branching microfluidic chip for digital quantitation (Fig. 1c). Target-liberated HRP@ZIF-8 nanoparticles are distributed and partitioned into individual microfluidic microwells, where they singly catalyze strong chemifluorescence signal enhancement for digital quantification. All fluidic actuations (i.e., nanoparticle partitioning, substrate introduction and microwell sealing) are powered through vacuum loading (Supplementary Fig. 1e).

As compared to conventional RNA detection approaches (e.g., PCR), which require multiple processing steps (e.g., reverse transcription and exponential target amplification) and are susceptible to variable amplification efficiencies (Supplementary Fig. 2), EZ-READ leverages highly catalytic transducers to achieve direct and reliable RNA detection (Fig. 1d). Specifically, through regenerative transduction, EZ-READ bypasses all steps of conventional PCR detection and achieves robust detection; through potent catalysis by the liberated HRP@ZIF-8 nanoparticles, the platform enables digital counting and reliable detection of RNA signatures. Motivated by its direct and accurate detection capabilities, we employed the EZ-READ platform to measure different RNA subtypes (e.g., long mRNA and short miRNA) found in clinical blood samples of GBM patients (Fig. 1e), and further developed a multi-step classification model to accurately diagnose patients and molecularly subtype GBM tumors (Supplementary Fig. 3).

### Regenerative RNA transduction
To achieve regenerative signal transduction, we first characterized and confirmed the composition of the prepared HRP@ZIF-8 nanoparticles (Supplementary Fig. 4a–c). As compared to pristine HRP, HRP@ZIF-8 showed enhanced enzyme activity as well as improved stability in complex environments. Specifically, the HRP@ZIF-8 nanoparticles demonstrated a characteristic redshift in their fluorescence spectrum (Fig. 2a), indicating a conformational change in the HRP active pocket[24]. Importantly, when measured against different substrate concentrations, HRP@ZIF-8 showed improved enzyme kinetics (Fig. 2b and Supplementary Fig. 4d); when fitted according to the Michaelis−Menten equation, the HRP@ZIF-8 reactions demonstrated a lower $K_M$, thereby confirming the nanoparticle's improved substrate affinity. All comparisons were matched in enzyme concentration, as verified through independent HRP formulations. In addition to demonstrating potent catalytic performance, the HRP@ZIF-8 nanoparticles also protected the embedded enzymes against environmental interferences (Fig. 2c and Supplementary Fig. 4e), thus

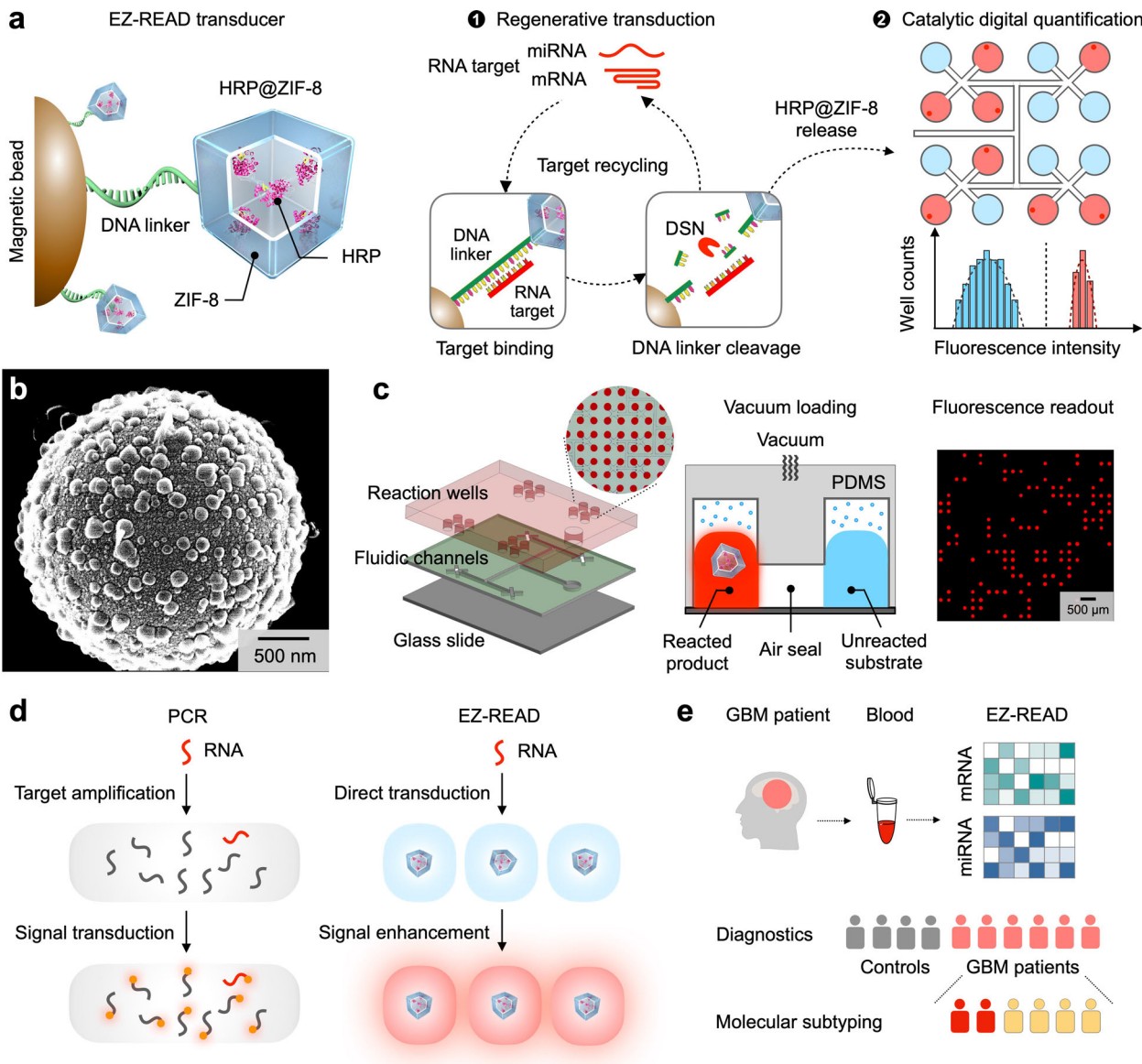

**Fig. 1 | EZ-READ platform for reliable profiling of circulating RNAs. a** EZ-READ technology. EZ-READ leverages an RNA-responsive transducer to regeneratively convert and catalytically enhance rare RNA targets. Each transducer carries catalytic nanoparticles – horseradish peroxidase (HRP) encapsulated in metal organic frameworks (ZIF-8) – via specific DNA probes. The scaffolded HRP@ZIF-8 nanoparticles thus bear strong catalytic activity and confer robust protection against environmental effects. During regenerative transduction, upon specific hybridization by RNA targets (e.g., miRNA and mRNA), the transducer activates directly, through selective cleavage of RNA-bound DNA linkers by duplex-specific nuclease (DSN), to release HRP@ZIF-8 nanoparticles in a target-recyclable manner. When partitioned within a fractal branching microfluidic chip, the released nanoparticles can singly catalyze strong chemifluorescence reaction to achieve digital quantification. **b** RNA-responsive transducer. Scanning electron micrograph of the assembled transducer confirmed high-density coverage by HRP@ZIF-8 nanoparticles. This experiment was repeated thrice independently with similar results. **c** Catalytic digital quantification. To exploit the potent activity of HRP@ZIF-8 nanoparticles for digital counting, we developed a fractal branching microfluidic chip with resistance-matched microwells to enable even nanoparticle partitioning and independent catalysis. Left: exploded view of the microfluidic chip. Middle: schematics of catalytic digital quantitation. HRP@ZIF-8 nanoparticles, reaction substate and air are sequentially introduced into the chip through vacuum loading to achieve independent catalysis. Right, fluorescence image of the reacted chip for digital counting. This experiment was repeated thrice independently with similar results. **d** Direct and reliable RNA detection by EZ-READ. As compared to conventional approaches (e.g., PCR) which require target amplification and are susceptible to variable amplification efficiencies, EZ-READ bypasses all steps of conventional PCR detection and achieves direct transduction and catalytic signal enhancement. **e** Clinical application. Leveraging its programmable detection, we applied EZ-READ to measure various RNA subtypes in clinical blood samples and developed multi-step classification models to accurately diagnose GBM patients and molecularly subtype the tumors.

enabling regenerative signal transduction (i.e., the incorporation of DSN and associated buffer) in a one-step reaction.

We next assembled the EZ-READ RNA-responsive transducer, by covalently linking the catalytic HRP@ZIF-8 nanoparticles to magnetic beads via specific DNA probes. In preparing the transducer, we coated the HRP@ZIF-8 nanoparticles with polyacrylic acid (PAA) and polyethylene glycol (PEG) spacers to improve DNA probe binding and RNA-induced nanoparticle release (Supplementary Fig. 4f, g). When treated with DSN to mediate cleavage of the HRP@ZIF-8 nanoparticles, the assembled transducer showed specific and amplified response to RNA targets, but not to DNA targets (Fig. 2d). Notably, as compared to sequence-specific DNA cleavage which recognizes dedicated cut

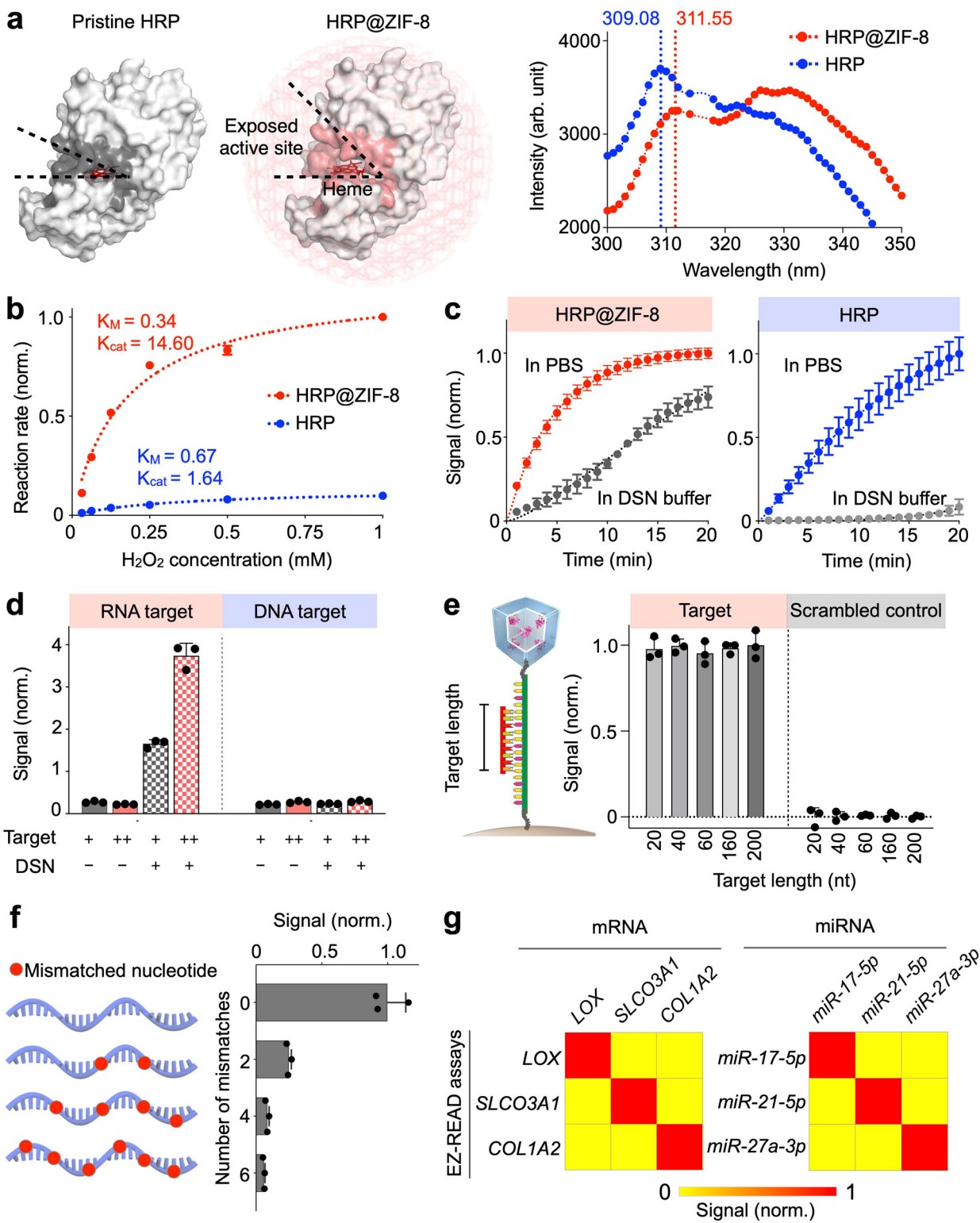

sites and generates distinct DNA products, the DSN system enables enhanced DNA cleavage and broad versatility against different target sequences. It recognizes RNA-DNA heteroduplexes and cleaves only the DNA strands in a random, repeated fashion to yield short DNA products (≤6 nt products[23], Supplementary Fig. 5a and Supplementary Table 1); these short DNA products can thus readily dissociate from the RNA target. Indeed, in the EZ-READ system, this effective DNA cleavage not only dissociates the attached HRP@ZIF-8 nanoparticles for signal liberation, but also regenerates the RNA targets for subsequent reactions, thereby enabling regenerative signal enhancement

(Supplementary Fig. 5b). Motivated by this effective transduction, we next evaluated the EZ-READ's assay programmability and specificity, respectively. To assess its ability to detect RNA targets of varying length (e.g., miRNA and mRNA), we designed transducers bearing different-sized DNA probes. When incubated with RNA targets of different length (20 to 200 nt), the EZ-READ technology generated strong, specific signals and showed minimal background signals with scrambled controls (Fig. 2e). When incubated with RNA targets bearing mismatches, the system further demonstrated good performance to distinguish rare mismatches (Fig. 2f and Supplementary Table 2). With

**Fig. 2 | RNA-responsive regenerative transducer. a** Enzyme conformational changes. As compared to pristine HRP, the ZIF-8 scaffolding (HRP@ZIF-8) configured the enzyme with a more accessible active site, as confirmed by a characteristic redshift in the HRP@ZIF-8 fluorescence spectrum. **b** Enhanced enzyme activity. Michaelis−Menten kinetics of pristine HRP and HRP@ZIF-8, when respectively treated with increasing substrate concentration, showed that the HRP@ZIF-8 has an improved enzyme substrate affinity and catalytic activity. **c** Robust protection against environmental effects. Both HRP@ZIF-8 and pristine HRP were incubated in duplex-specific nuclease (DSN) buffer. The ZIF-8 encapsulation protected the embedded enzymes and preserved their activity from complex environments. Pristine HRP, however, showed negligible residual activity in DSN buffer. This robust performance by HRP@ZIF-8 enabled the incorporation of DSN and its associated buffer to achieve regenerative transduction in a one-step reaction. **d** Regenerative transduction of RNA target. EZ-READ showed an enhanced

response to RNA target, but not to DNA target. When incubated with increasing amounts of RNA target, only in the presence of DSN, EZ-READ showed reflective signals. **e** Programmability for different RNA subtypes. EZ-READ transducers were designed to carry different-sized DNA probes. The transducers generated strong signals when incubated with specific RNA targets of different lengths (20–200 nt), and showed minimal background signals with scrambled controls. **f** Mismatch specificity of EZ-READ assay. When incubated with synthetic RNA targets bearing varying number of mismatches, the system demonstrated good performance to distinguish rare mismatches. **g** Specificity of EZ-READ assays. Assays were developed for both GBM-associated mRNAs (e.g., *LOX*, *SLCO3A1*, *COL1A2*) and miRNAs (e.g., *miRNA-17-5p*, *miRNA-21-5p*, *miRNA-27a-3p*). Heat map signals were assay (row) normalized. All measurements were performed in triplicate ($n = 3$ independent experiments), and the data are displayed as mean ± s.d. in (**b**–**f**) and as mean in (**g**). Source data are provided as a Source Data file.

this, we developed EZ-READ transducers for various GBM-associated RNA markers (e.g., mRNA: *LOX*, *SLCO3A1*, *COL1A2*, miRNA: *17-5p*, *21-5p*, *27a-3p*)[25,26] and confirmed their detection specificity (Fig. 2g and Supplementary Table 2).

## Catalytic digital EZ-READ for reflective target quantification

To accurately measure the regenerative transduction (i.e., the amount of HRP@ZIF-8 particles released), we developed a microfluidic platform to enable catalytic digital quantitation. The platform comprises a fractal branching array of microwells to partition and compartmentalize individual HRP@ZIF-8 particles, so as to exploit their strong catalytic activity for digital chemifluorescence detection (Fig. 3a). As compared to conventional microfluidic designs, which fill microwells sequentially and result in uneven partitioning (Supplementary Fig. 6a), the fractal branching configuration ensures equal fluidic resistance to individual microwells to enable even and controllable microwell filling (e.g., with HRP@ZIF-8 particles). We further optimized the fractal branching angle to maximize the microwell density in the chip (Supplementary Fig. 6b). With six layers of fractal branching, the main fluidic channel is repeatedly divided and subdivided into thousands of branches (Supplementary Fig. 6c). By terminally connecting each branch to four microwells, the current prototype houses a total of 4096 microwells in a 2.5 cm × 2.5 cm footprint (Supplementary Fig. 6d). To achieve catalytic digital quantitation, we sequentially loaded the microfluidic platform with HRP@ZIF-8 particles, chemifluorescence substrate, and air, to achieve stepwise reactions: (1) HRP@ZIF-8 partitioning into individual microwells, (2) substrate introduction to respective microwells, and (3) sealing of individual reaction chambers for independent catalysis and digital counting (Supplementary Movie 1). All fluidic actuations were powered by vacuum charging and microwell evaporation was minimized during the assay (Supplementary Fig. 7a, b). To optimize these stepwise reactions, we monitored the liquid occupancy in individual microwells to ensure loading efficiency and reaction completeness (Fig. 3b, Supplementary Fig. 7c and Supplementary Movie 2). Experimental validation further confirmed even particle partitioning and independent chemifluorescence reactions (Supplementary Fig. 7d) to accurately quantify the concentration of input HRP@ZIF-8 nanoparticles (Supplementary Fig. 8).

We next evaluated the performance of the integrated EZ-READ platform for direct RNA quantification. Employing the developed platform, we first performed a series of titration experiments with known amounts of RNA targets (e.g., mRNA and miRNA). RNA strands were incubated with EZ-READ transducers to release HRP@ZIF-8 particles (Supplementary Table 2). The liberated particles, evenly partitioned within microfluidic microwells, generated strong and representative chemifluorescence signals (Fig. 3c). The number of fluorescent microwells (i.e., positive wells) (Supplementary Fig. 9a) not only strongly correlated to nanoparticle occupancy, but also accurately determined input RNA amounts (Fig. 3d). When used to measure RNA target copies, the EZ-READ could be applied to measure a wide

range of RNA target concentrations and establish linear calibration for target quantitation (Supplementary Fig. 9b, c). In particular, for short miRNA detection, while gold standard RT-qPCR required extensive preparation steps to elongate and detect short miRNA (i.e., RNA extraction, reverse transcription and pre-amplification) (Supplementary Fig. 9d), the EZ-READ platform achieved direct and sensitive miRNA detection, establishing a limit of detection of 9 copies of miRNA (Fig. 3e). Specifically, the EZ-READ platform bypassed all the above-mentioned preparation steps of RT-qPCR for short miRNA analysis and enabled direct measurements in minimally processed sample lysates (Supplementary Fig. 10a). Through analytical evaluation, we next confirmed the technology robustness, when measurements were performed on different chips and by different users (Supplementary Fig. 10b). We finally assessed the accuracy of the EZ-READ platform to detect input RNA signatures. Using transducers against a clinically relevant mutation (i.e., *IDH1 R132H* mutation in brain tumors) (Supplementary Table 2), in patient tumor tissues as well as paired plasma samples with known IDH1 mutation status, we performed the EZ-READ measurements and established accurate tumor classification (overall classification accuracy ≥90%) (Supplementary Fig. 10c). Finally, for measurement of complex ratiometric RNA signatures, we found that while RT-qPCR showed poor correlations to input RNA ratios, likely due to its extensive sample processing and susceptibility to variable amplification efficiencies, the EZ-READ platform measured directly and accurately to reflect the input RNA signatures (Fig. 3f).

## Selection of RNA markers for GBM subtyping

To clinically apply the EZ-READ platform for GBM subtyping, we first selected RNA markers for GBM diagnosis and subtype classification. Based on previous studies[25,26], we chose 22 mRNA markers and 10 miRNA markers and evaluated these markers through cell lines as well as clinical specimens. For cell line characterization, we cultured six primary GBM cell lines (NNI-22, NNI-24, NNI-32, NNI-11, GLI36vIII and SKMG3) and isolated their EVs. Through molecular characterization, we demonstrated that the vesicles have a mean diameter of ~130 nm and express characteristic EV protein markers and RNA composition (Supplementary Fig. 11). We further measured the RNA contents of the cells and their derived EVs (Fig. 4a). The mRNA and miRNA profiles between the EVs and parent cells correlated well (Supplementary Fig. 12a), indicating that EVs are reflective of their parent cells and have the potential to be used as a surrogate biomarker for cells.

We further evaluated the RNA markers in clinical plasma samples and primary tumor tissues from GBM patients, to assess the markers' expression and correlation to tissue subtypes. First, we measured these RNA markers in paired plasma and tumor tissues. Interestingly, the clinical plasma samples showed low expression of many mRNA markers (Fig. 4b), as supported by the decreased abundance of mRNA in EVs (Supplementary Fig. 11d), and demonstrated a different RNA profile to that of the primary tissues (Supplementary Fig. 12b).

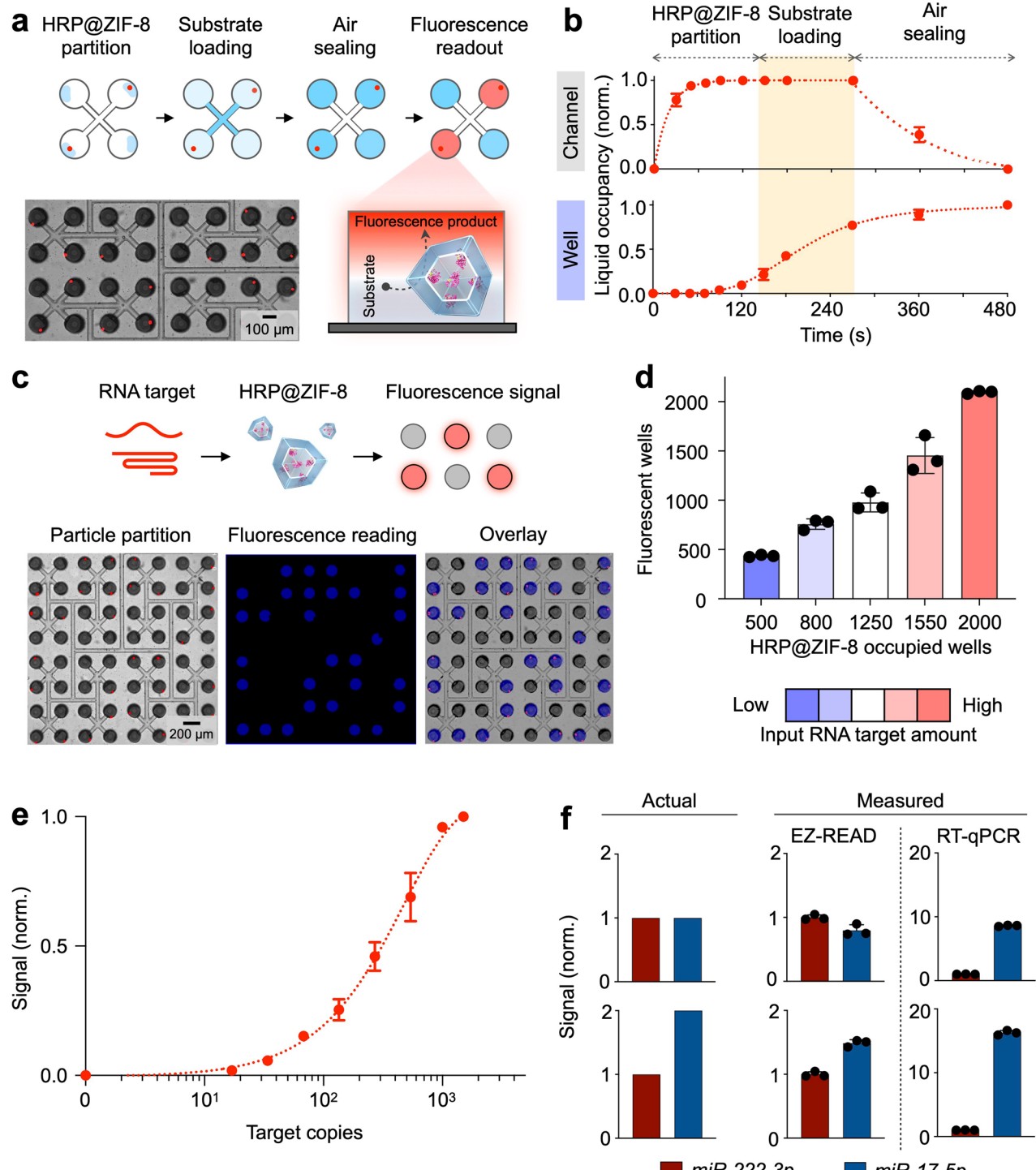

**Fig. 3 | Catalytic digital quantification of RNA targets. a** Schematics for catalytic digital quantification. We developed a microfluidic platform to leverage the strong catalytic activity of HRP@ZIF-8 nanoparticles for digital detection. Specifically, the platform comprises a fractal branching array of resistance-matched microwells to achieve stepwise loading of HRP@ZIF-8 particles, reaction substrate, and air sealant, so as to achieve even particle compartmentalization and independent catalytic readout. Left inset shows a photograph of microfluidic chip, when loaded with fluorescently labeled HRP@ZIF-8 particles. **b** Optimization of stepwise reactions. We monitored the liquid occupancy in the microfluidic channels and microwells to ensure loading efficiency and reaction completeness. **c** EZ-READ detection of RNA. RNA targets were directly incubated with EZ-READ transducers to regeneratively release HRP@ZIF-8 nanoparticles. The released particles were evenly partitioned within microfluidic microwells, and reacted to generate chemifluorescence signals. Using fluorescently labeled HRP@ZIF-8 nanoparticles, we confirmed independent

chemifluorescence reactions that closely matched nanoparticle occupancies. **d** Performance validation for RNA quantification. We treated EZ-READ with different amounts of input RNA. The resultant number of fluorescent microwells (i.e., positive wells) not only strongly correlated to nanoparticle occupancy, but also accurately measured input RNA amounts. **e** Sensitivity of EZ-READ assay. The limit of detection (LOD) was determined by titrating a known amount of short miRNA target (*miR-222-3p*) and counting the number of fluorescent wells. EZ-READ showed a limit of detection of 9 copies of miRNA target. The LOD is defined as 3 × s.d. of a no-target control. **f** Reflective quantification of RNA signatures. When treated with a series of samples with known input RNA ratios, EZ-READ accurately measured the input RNA signatures while RT-qPCR showed poor correlation. All measurements were performed in triplicate (*n* = 3 independent experiments), and the data are displayed as mean ± s.d. in (**b**) and (**d**–**f**). Source data are provided as a Source Data file.

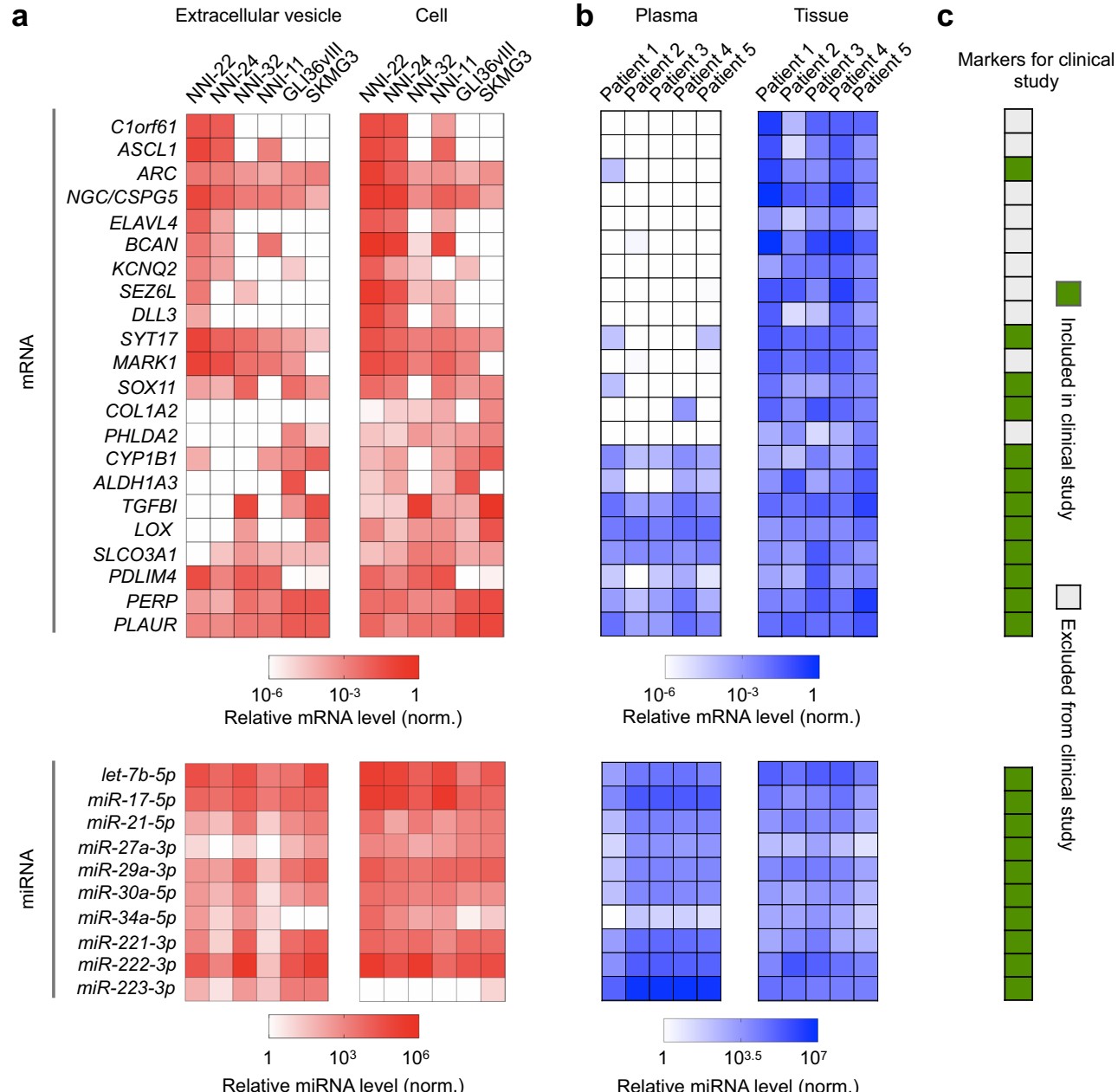

**Fig. 4 | Molecular profiling of GBM samples and marker selection. a** RNA analysis of GBM EVs and their parent cell lines. We obtained cell pellets and isolated EVs from six GBM lines. EV and cellular mRNA and miRNA levels showed good correlation, indicating that EVs are reflective of their parent cells and have the potential to be used as a surrogate biomarker for cells. **b** RNA analysis of matched plasma-derived EVs and tumor tissues obtained from GBM patients. Plasma and tumor miRNA levels were correlated while their mRNA levels were not. The expression levels of mRNA and miRNA were measured with RT-qPCR. mRNA levels were normalized to intrinsic GAPDH and miRNA levels normalized to total RNA amount input. **c** Markers selected for inclusion in subsequent clinical study. mRNA markers with plasma expression and all miRNA markers were included in the clinical study. All measurements were performed in triplicate ($n = 3$ independent experiments), and the data are displayed as mean. Source data are provided as a Source Data file.

We attribute this to heterogeneous EV composition in plasma; EVs are released by multiple cell types into the circulation[14]. We therefore selected mRNA targets that were expressed in plasma and all miRNA targets for our subsequent clinical classification study (Fig. 4c and Supplementary Fig. 12c). Next, we independently verified these selected RNA targets using additional GBM tumor tissues of different GI subtypes (Supplementary Fig. 13a, b). Specifically, by testing the markers with established GI signature genes[7], we performed 5-fold cross-validation analysis and showed that the selected mRNA targets contribute to the variability in GI subtype classification (Supplementary Fig. 13c). These targets were not only expressed in plasma but also strongly associated with different GI subtypes, thereby supporting their potential application for blood-based disease subtyping.

## Clinical analysis of GBM diagnosis and subtyping

Using the selected RNA markers, we finally conducted a clinical feasibility study to evaluate the ability of the EZ-READ platform for blood-based RNA profiling of GBM. GBM is grade 4 glioma and can be transcriptionally characterized by three subtypes with differential prognosis and treatment response[5,6]. In this study, we aim to determine (1) if the EZ-READ platform could be applied to clinical plasma samples for diverse profiling of different RNA molecules (e.g., miRNA and mRNA),

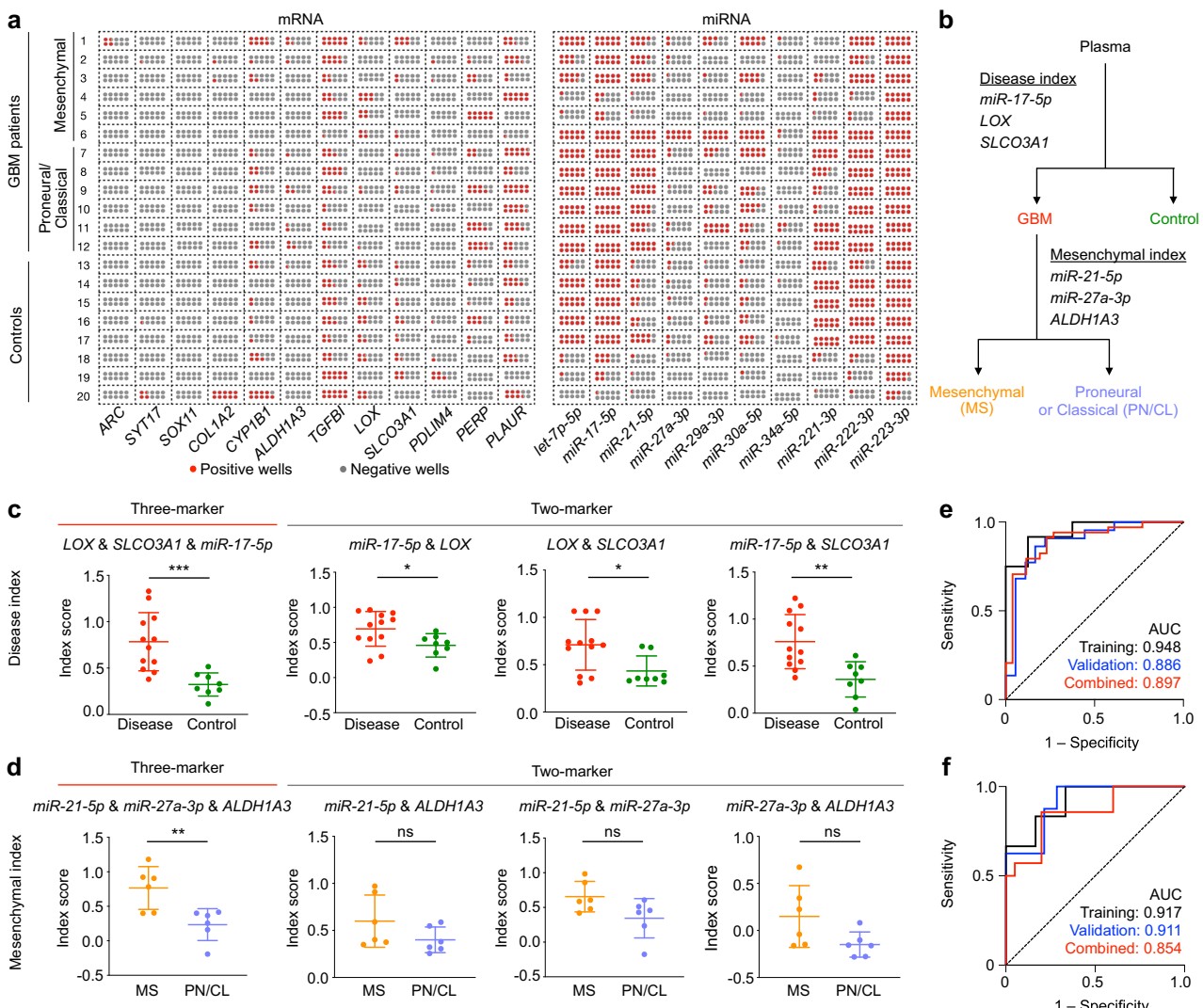

**Fig. 5 | Clinical validation of EZ-READ. a** EZ-READ analysis of mRNA and miRNA markers in a training clinical cohort. The training set consisted of *n* = 20 subjects; 12 disease (6 mesenchymal and 6 proneural/classical) and 8 control. Measurements were performed in triplicate (*n* = 3 independent experiments). **b** Multi-layer decision model for stratification of plasma samples. Disease index was first obtained with multiple linear regression to categorize samples into Disease or Control. For the samples classified as Disease, the Mesenchymal index was generated with multiple linear regression to distinguish the tumor subtype from others. **c** Disease classification of the training cohort. The selected three-marker model outperformed constituent two-marker models at classifying the disease and control samples. *n* = 20 subjects, measurements were performed in triplicate (*n* = 3 independent experiments). Data are displayed as mean ± s.d. ns = not significant,

*P < 0.05, **P ≤ 0.01, and ***P ≤ 0.001, two-sided Student's t test. For *LOX* & *SLCO3A1* & *miR-17-5p*, P = 0.0010. **d** Mesenchymal classification of the training cohort. The selected three-marker model outperformed constituent two-marker models at classifying the mesenchymal samples. *n* = 12 subjects, measurements were performed in triplicate (*n* = 3 independent experiments). Data are displayed as mean ± s.d. ns = not significant, *P < 0.05, **P ≤ 0.01, and ***P ≤ 0.001, two-sided Student's t test. For *miR-21-5p* & *miR-27a-3p* & *ALDH1A3*, P = 0.0071. **e–f** Receiver operating characteristic (ROC) curves of the Disease regression model (**e**) and the Mesenchymal regression model (**f**) on the training, validation and combined patient cohorts, respectively. The independent validation cohort consisted of *n* = 40 subjects; 22 disease (8 mesenchymal and 14 proneural/classical) and 18 control. Source data are provided as a Source Data file.

and (2) the accuracy of the EZ-READ RNA signatures in distinguishing and subtyping GBM patients.

We obtained blood samples from GBM patients (*n* = 34) and control subjects (*n* = 26), and randomized them into two representative cohorts (i.e., training and validation, Supplementary Table 3) to independently evaluate the robustness of EZ-READ measurements (Supplementary Fig. 14). All GBM subjects were clinically characterized, through RNA sequencing of the primary tumor tissue for disease subtyping, and showed indistinguishable plasma EV characteristics across groups (Supplementary Fig. 15). In the training cohort that comprises 12 disease samples and 8 control plasma samples, we first employed the EZ-READ platform to measure the selected RNA biomarkers (Fig. 5a). Leveraging the measured RNA abundance, we constructed a multi-layer decision model that could

(1) differentiate between disease and control samples and (2) distinguish the mesenchymal subtype, the most aggressive GBM subtype with the worst prognosis (Fig. 5b). In developing this sequential model, we characterized the performance of individual markers through receiver operating characteristic (ROC) curve analysis (Supplementary Fig. 16a). To improve the classification accuracy, we further performed stepwise linear regression to identify multi-marker panels: disease index comprising *miR-17-5p*, *LOX* mRNA and *SLCO3A1* mRNA to differentiate patients from controls (Fig. 5c), and mesenchymal index comprising *miR-21-5p*, *miR-27a-3p* and *ALDH1A3* mRNA to distinguish the mesenchymal subtype from other disease samples (Fig. 5d). To evaluate the performance of the EZ-READ classification model, we further assessed the technology in an independent validation cohort comprising 22 disease and 18 control patient

samples. As compared to the training cohort, the developed model indices showed comparable performance in the validation cohort and achieved accurate classification (Fig. 5e, f, AUC = 0.897 and 0.854 for the combined cohort, respectively). The multi-step decision model outperformed the single-step multinomial regression and showed superior classification accuracy (Supplementary Fig. 16b). Furthermore, the EZ-READ classification model showed better performance than conventional RT-qPCR (Supplementary Fig. 16c), likely due to its improved robustness and reliable measurements.

## Discussion

Minimally-invasive approaches that can molecularly characterize GBM can bring forth new clinical opportunities for personalized treatment. Recent studies have shown that circulating RNAs (e.g., mRNA and miRNA) are an attractive biomarker for blood-based assessment of GBM[10–12]. Nevertheless, current analytical technologies face limitations to reliably measure these scarce and diverse circulating RNAs, especially due to the approaches' complex sample processing and vulnerability to amplification variabilities. To address these challenges, we have developed an analytical platform to directly transduce and catalytically enhance RNA signals, thereby enabling accurate and digital quantification of diverse circulating RNAs for GBM profiling.

The EZ-READ platform leverages the following components to achieve regenerative transduction and digital quantitation: (1) it employs a catalytic transducer that liberates highly potent HRP@ZIF-8 nanoparticles to individually catalyze digital quantification; (2) it utilizes DSN to achieve effective RNA target recycling. The enzyme recognizes RNA-DNA heteroduplexes (instead of sequence-specific cut sites) and cleaves only the DNA strands to yield very short fragments; this mechanism enables not only regenerative transduction but also programmable assay expansion; (3) its microfluidic platform features a fractal branching network of microchannels. This design ensures equal fluidic resistance leading to individual microwells, enables even reagent loading and independent reactions for digital counting, without needing sophisticated liquid handlers, and can be readily scaled to increase the number of reaction microwells. As compared to conventional RNA detection approaches (e.g., hybridization and target amplification methods), the EZ-READ platform is thus well-suited for direct and accurate measurement of diverse RNA signatures (Supplementary Table 4). With respect to traditional hybridization-based assays (e.g., fluorescence resonance energy transfer/FRET-based molecular beacons) which rely on target binding to induce fluorescence signal change and have limited sensitivity, the EZ-READ achieves sensitive and accurate digital quantification. With respect to target amplification assays (e.g., RT-qPCR) which require extensive sample processing (e.g., reverse transcription and target amplification), the EZ-READ bypasses all steps of conventional PCR to enable direct and robust detection of different-sized RNA subtypes (miRNA and mRNA).

The scientific and clinical applications of the developed technology are potentially broad. With its programmable detection and robust performance in minimally processed samples, the technology could theoretically be expanded to measure other biomarkers, through the incorporation of alternative probes[27,28], nanomaterials[29,30] and responsive cascades[31–33]. Likewise, the integration of advanced microfluidics[34–36] could not only enhance the technology's analytical performance, but also scale its multiplexing and throughput capabilities for biomarker discovery and evaluation studies (e.g., measurements of *TERT* promoter mutations and in larger patient cohorts). Clinically, minimally-invasive approaches to accurately characterize disease status and molecular subtypes could plausibly lead to advances in personalized treatment, in GBM and beyond[2,37]. While the current study measures only circulating RNAs, combinatorial analysis of additional circulating markers (e.g., blood-borne proteins and metabolites) could refine existing biomarker signatures and develop valuable composites[38,39]. With the EZ-READ's programmable detection and

rapid operation, we further envision that the technology could be applied to discover and translate additional biomarker signatures in a spectrum of diseases (e.g., cancers, cardiovascular diseases and neurological diseases), using various easily accessible bodily fluids (e.g., blood, ascites, urine and saliva), to facilitate real-time monitoring and guide personalized treatment[40,41]. Through additional technical innovations, such as on-chip processing[42] and device automation[43,44], the technology could be expanded to accelerate large cohort clinical validations.

Finally, to enable translation, regulatory clearance and commercialization strategy are critical for the technology's maturation and adoption[45,46]. For regulatory approval, the EZ-READ platform could either be deployed as a laboratory-developed test or an in vitro diagnostic (IVD) test and this decision has extensive regulatory implications[47,48]. Laboratory-developed tests can only be used in dedicated Clinically Laboratory Improvement Amendments (CLIA)-certified laboratories and are thus exempted from extensive regulatory oversight. In comparison, IVD tests require much more stringent regulatory clearance (e.g., analytical validation, clinical validation and clinical utility) for distributed use as standalone assays. To facilitate commercialization, multifaceted factors need to be considered upfront and progressively. These include clinical use cases, scale-up manufacturing, quality management as well as reimbursement model, amongst others, and require extensive collaborations across multiple stakeholders (e.g., scientists, clinicians, commercial developers and policymakers).

## Methods

### Ethical statement

This study was approved by the National University of Singapore Institutional Review Board (NUS-IRB no. 2021-152). De-identified clinical specimens were obtained with informed consent from the National Neuroscience Institute Tissue Bank (application no. SBRSA2019/002) in accordance with the SingHealth Centralized Institutional Review Board.

### HRP@ZIF-8 nanoparticle complexes synthesis and characterization

2-Methylimidazole (2.5 M, Sigma-Aldrich) was prepared in water and mixed with streptavidin-HRP (5 μg/ml, Thermo Scientific), polyacrylic acid (0.2 M, Sigma-Aldrich) and zinc nitrate hexahydrate (0.5 M, Sigma-Aldrich) at 25 °C under stirring condition. After incubation for 30 min, the suspension was centrifuged at 500 g and washed with phosphate-buffered saline (PBS). The as-synthesized hybrid HRP@ZIF-8 nanoparticles were characterized for their hydrodynamic diameter and zeta potential through dynamic light scattering analysis (Zetasizer Nano ZSP, Malvern). $3 \times 14$ measurement runs were performed at room temperature. Z-average diameter, zeta potential and polydispersity were analyzed. For every measurement, the auto-correlation function and polydispersity index were monitored to ensure sample quality for size and surface charge determination. Powder X-ray diffraction was performed in the 2θ range 5–50 ° at a scanning rate of 2 °/min on an X-ray diffractometer (Bruker D8 Advanced) with a Cu-Kα radiation at 40 kV and 40 mA. Fourier transform infrared spectroscopy (Bruker) was performed at an attenuated total reflectance unit in the range of 500–3500 cm$^{-1}$.

### Enzyme characterization

To characterize enzyme conformational changes, we measured its fluorescence emission spectrum with excitation at 280 nm for tryptophan and curve-fitted the peaks (Tecan, SparkControl v2.1). A typical redshift of tryptophan emission spectrum indicates positive HRP structural changes, which helps to expose the HRP heme group and increase its substrate affinity[24]. Enzyme kinetics studies were performed through a commercial kit (QuantaBlu Fluorogenic Peroxidase Substrate Kit, Thermo Scientific). Briefly, working solutions were

prepared by mixing QuantaBlu substrate solution with peroxide solutions of different concentration, at a fixed volume ratio of 9:1. HRP@ZIF-8 was then mixed with different working solutions. For each reaction, fluorescence intensity (excitation/emission: 325 nm/420 nm) was monitored for 20 min and time-fitted to determine the initial reaction rate (time = 0 s). Finally, the Michaelis–Menten equation was applied to determine the $K_M$ and $K_{cat}$.

## RNA-responsive transducer assembly

To assemble the RNA-responsive transducer, carboxyl magnetic beads (3 μm, Spherotech) were activated through carbodimide cross linking, in a mixture of excess NHS/EDC dissolved in MES buffer (pH 4.7) for 15 min. After rinsing, the activated magnetic beads were incubated with heterobifunctionalized polyethylene glycol (PEG) bearing terminal amino and maleimide group (0.1 mM, JenKem). Separately, DNA probes with 5′ thiol and 3′ amino group (30 μM, Integrated DNA Technologies) were activated with reducing tris (2-carboxyethyl) phosphine and recovered through filtration. The activated DNA probes were subsequently added to the prepared maleimide-functionalized magnetic beads. This reaction was washed in excess PBS, before incubation with another heterobifunctionalized PEG bearing terminal NHS and biotin group (1 mM, Thermo Scientific). Finally, the DNA-functionalized magnetic beads were added to excess HRP@ZIF-8 nanoparticles, washed in PBS and stored at 4 °C for subsequent use.

## Regenerative signal transduction

To evaluate the assembled RNA-responsive transducers, we prepared FITC-labeled HRP@ZIF-8 nanoparticles and functionalized these onto magnetic beads. The assembled transducers were incubated with RNA targets in a reaction buffer (50 mM Tris-HCl, pH 8.0, 5 mM MgCl₂, 1 mM DTT, 6 U/ml RNase inhibitor) containing 0.5 U of duplex-specific nuclease (DSN, Evrogen). After magnetic pulldown, the reaction supernatant containing released HRP@ZIF-8 nanoparticles was collected and measured through fluorescence analysis. To study the kinetics of HRP@ZIF-8 release, we monitored the fluorescence intensity of the supernatant over time and analyzed the signal difference between reactions incubated with high (10 pmol) and low (3 pmol) RNA target amounts. To evaluate the versatility and specificity of the transducers, they were mixed with RNA targets with varying length (20 to 200 nt) or bearing different number of mismatches.

## Validation of DSN cleavage mechanism

To demonstrate the mechanism of DSN cleavage, we prepared RNA-DNA heteroduplexes by mixing an equal molar ratio of RNA target and DNA probe in a reaction buffer (50 mM Tris-HCl, pH 8.0, 5 mM MgCl₂, 1 mM DTT) and incubated the mixture at 80 °C for 15 min followed by cooling to 4 °C. Annealed RNA-DNA heteroduplexes were then subjected to DSN cleavage by incubation at 60 °C for 7 min followed by cooling to 4 °C. For comparison, we subjected the annealed RNA-DNA heteroduplexes to BstNI digestion[49] (cut site CC/WGG, New England Biolabs) by incubation at 60 °C for 1 h followed by cooling to 4 °C. The DNA samples before and after enzyme digestion were extracted (Qiaquick, Qiagen) and analyzed by native polyacrylamide gel electrophoresis (PAGE). DNA samples were diluted with appropriate amounts of 6X loading dye (Promega) and run on a 10% polyacrylamide gel with TAE buffer (Vivantis) at 150 V. The gel was stained with 1X SYBR Gold (Invitrogen) in TAE buffer for 30 min before being imaged using a iBright FL1500 Imaging System (Thermo Scientific).

To evaluate the DSN efficiency in the EZ-READ platform, we prepared free-floating RNA-DNA heteroduplexes, with FRET-based DNA probes, as described above. We treated this gold standard mixture with DSN and measured the amount of cleaved fluorescent probes to determine the maximal amplification efficiency (100%). In comparison, to characterize the EZ-READ platform, equal concentration of magnetic bead-immobilized RNA-DNA heteroduplexes were applied for

DSN cleavage. The EZ-READ amplification efficiency was normalized to that of the free-floating gold standard and measured 93.4%. The reduced efficiency is likely due to decreased enzyme accessibility due to probe immobilization on the beads.

## Design of digital microfluidics

To develop a microfluidic platform for catalytic digital characterization of HRP@ZIF-8 released, we first performed computational fluid dynamics analysis (COMSOL Multiphysics) to establish the flow profile and optimize the channel design. In simulation, the fluid material inside the channel was set as water, and an incompressible flow model was applied to the fluid. The numerical simulations were conducted considering no-slip wall condition and zero pressure in the outlet section. Velocity of 0.1 m/s at the inlet section and the laminar flow module with a steady-state study were applied. The environmental temperature was set at T = 293.15 K. In addition, the particle tracing module considering the drag force on the particles was used to simulate the particle trajectory within the microwells.

## Fabrication and preparation of digital chips

Standard soft lithography was used for fabricating the microfluidic chips. We used SU-8 negative resist (SU8-50 and SU8-100, Microchem) to prepare the mold. Briefly, to form the connecting fluidic channel, the photoresist SU8-50 was spin-coated onto a Si wafer at 3000 rpm for 30 s, and baked at 65 °C and 95 °C for 5 min and 15 min, respectively. After UV light exposure, the resist was baked again before being developed. To fabricate the second layer comprising digital microwells, the photoresist SU8-100 was spin-coated onto a Si wafer at 1000 rpm for 30 s, degassed for 30 min, and baked at 65 °C and 95 °C for 30 min and 90 min, respectively. The photomask was properly aligned to the marker on the first layer before exposing and developing the resist. The fabricated fluidic channels and digital wells were 40 μm and 250 μm in height, respectively. The developed mold was chemically treated with trichlorosilane vapor inside a desiccator for 15 min and baked at 150 °C for 10 min to evaporate the excessive silane before subsequent use. Polydimethylsiloxane polymer (PDMS) and cross-linker were mixed in a 10:1 ratio and cast onto the SU-8 mold. After curing at 65 °C for 4 h, the PDMS layer was cut from the mold and plasma-treated before assembly onto the glass slide. The inlet was made with a 4-mm biopsy punch for sample loading. After fabrication, to achieve automatic sample loading, we first performed vacuum charging of the fabricated digital chips. Chips were sealed using MicroAmp optical adhesive film (Applied Biosystems) and incubated for 2 h at 10 mbar before sample loading.

## EZ-READ assay workflow

We employed the prepared EZ-READ microfluidic platform to achieve catalytic digital quantification of RNA targets. Specifically, 3 μl of sample containing RNA targets was first mixed with 8 μl of RNA-responsive transducers to trigger HRP@ZIF-8 nanoparticle release. After 7 min of incubation at 60 °C, 5 μl of the HRP@ZIF-8 supernatant was added to a vacuum-charged microfluidic chip. At an average flow rate of 10 μl/min, the released HRP@ZIF-8 particles were automatically and evenly loaded into individual microwells within 2.5 min. To catalytically quantify individual HRP@ZIF-8 nanoparticles, we incorporated a commercial kit (QuantaRed Enhanced Chemifluorescent HRP Substrate, Thermo Scientific) that produces HRP-dependent fluorescence signal, through the generation of esorufin, a soluble and fluorescent product with excitation/emission maxima of 570/585 nm. Briefly, 20 μl of QuantaRed substrate was introduced to the microfluidic chip in 5.5 min. Subsequently, air was introduced into the microfluidic channel to seal off individual microwells, to enable independent catalytic reactions and reduce cross contamination among the microwells. The catalysis was allowed to proceed for 1 min, before fluorescence data acquisition. All experiments were performed with

sample-matched controls, where respective samples were incubated with transducers against *Arabidopsis thaliana ath-miR159a*, a target absent in human samples.

## EZ-READ signal processing

We captured fluorescence and bright-field images of the EZ-READ microwells with an inverted fluorescence microscope (DMi8 Leica). Image acquisition was performed with LAS X software (v.3.6.123246) and image analysis with ImageJ (v.1.53k). Final images were merged from 81 individual images, each collected at 5× zoom and 16 bits, to achieve an overall 16377 × 16377 pixels and final area of 21.28 × 21.28 mm$^2$. All fluorescence images were captured in greyscale and automatically processed (pseudo-colored red or blue) using the LAS X software according to the greyscale intensity. For digital quantitation, the greyscale images were used directly for ImageJ analysis (particle counting function) to acquire a distribution of fluorescence intensity in individual wells. We used this automated analysis to establish threshold and determine the number of positive wells. Image settings were applied consistently to all images of sample and control groups. The final signal for each sample was determined by subtracting the number of positive wells measured in its sample-matched control.

## Cell culture

Human glioblastoma (GBM) cell lines GLI36vIII and SKMG3 were provided by Dr. Timothy Chan, Memorial Sloan-Kettering Cancer Center, and grown in Dulbecco's Modified Essential Medium (DMEM, Gibco), supplemented with 10% fetal bovine serum (FBS). Human glioma-propagating cell lines NNI-11, NNI-22, NNI-32 and NNI-24 were generated from primary GBM samples, provided by National Neuroscience Institute (NNI) and cultured as neurospheres[50] in a serum-free 3:1 mix of Dulbecco's modified Eagle's medium (DMEM, Sigma-Aldrich) and Ham's F-12 Nutrient Mixture (F12, Gibco) supplemented with basic fibroblast growth factor (bFGF, Peprotech Inc., 20 ng/ml), epidermal growth factor (EGF, Peprotech Inc., 20 ng/ml), heparin (Sigma-Aldrich, 5 µg/ml), and serum-free supplement (B27, 1x, Gibco). All media were supplemented with penicillin-streptomycin. The cultures were incubated at 37 °C in a water-saturated atmosphere containing 5% CO$_2$. All cell lines were tested and free of mycoplasma contamination (MycoAlert Mycoplasma Detection Kit, Lonza, LT07-418).

## Extracellular vesicle isolation and quantification

Cells were cultured in vesicle-depleted medium (with 5% depleted FBS) for 48 h before vesicle collection. All media containing vesicles were filtered through a 0.2-µm membrane filter (regenerated cellulose, Millipore), isolated by differential centrifugation (first at 10,000 g and subsequently at 100,000 g), and used for molecular analysis. For independent quantification of vesicle concentration, we used the nanoparticle tracking analysis (NTA v3.3) system (NS300, Nanosight). Vesicle concentrations and diameters were adjusted to obtain ~50 vesicles in the field of view to achieve optimal counting. All NTA measurements were done with identical system settings for consistency.

## Clinical samples

De-identified GBM tumor specimens were obtained with informed consent from the National Neuroscience Institute Tissue Bank (application no. SBRSA2019/002). Tumor tissues were obtained fresh at the time of surgical resection and processed immediately. Under sterile condition, tumor tissues were subdivided into small pieces and processed respectively for RNA sequencing and immunohistochemical staining. Specifically, for RNA analysis, tumor pieces were snap-frozen in cryovials immersed in liquid nitrogen and stored at −150 °C. For immunohistochemical studies, tumor pieces were either fixed in 4% paraformaldehyde at 4 °C, dehydrated and embedded in paraffin, or incubated in 30% sucrose at 4 °C, followed by quick freezing, within a plastic mold, in optimal cutting temperature medium on dry ice. The tissue blocks were stored at −80 °C before subsequent analysis. For plasma samples, venous blood was drawn prior to tumor resection in EDTA tubes and processed immediately. Blood samples were centrifuged for 10 min at 400 g (4 °C). The recovered plasma was centrifuged again for 10 min at 1100 g (4 °C). Plasma samples were stored at −80 °C before use. All EZ-READ measurements were performed blinded from clinical diagnoses.

## Direct EZ-READ detection in biological lysates

For the evaluation of EZ-READ performance in lysed biological samples, we prepared GLI36vIII lysates using various lysis protocols[28] and applied EZ-READ for direct measurement, without RNA extraction. For chemical lysis, we prepared lysis buffers containing varying amounts of Triton X-100 and sodium dodecyl sulfate (SDS) and incubated these with the cell pellets at room temperature. For sonication, cell pellets were resuspended in PBS, and sonicated at 80 °C for 5 min (Elma). For plasma lysis, plasma samples were incubated with Triton X-100 and SDS for 5 min, at a final concentration of 1% Triton X-100 and 0.1% SDS. EZ-READ measurements were performed in these lysates directly, using assay procedures as described above.

## RNA extraction and quantification

As a gold standard, biological samples were harvested in TRIzol reagent for RNA extraction using a commercially available kit (miR-Neasy, Qiagen). RNA was extracted from the lysates and treated with DNase I (Qiagen) to remove DNA contamination, per manufacturer's protocol. Extracted RNA was quantified using a NanoDrop spectrophotometer (Thermo Scientific). To evaluate the size distribution of extracted RNA, purified RNA samples were evaluated with 2100 Bioanalyzer (Agilent) using a RNA Pico Chip.

## RT-qPCR analysis

For RT-qPCR, extracted mRNA and miRNA were first reverse-transcribed to generate first-strand cDNA (High-Capacity cDNA Reverse Transcription Kit and TaqMan MicroRNA Reverse Transcription Kit, Applied Biosystems). cDNA was pre-amplified where necessary (TaqMan PreAmp Master Mix, Applied Biosystems) before qPCR. All qPCR reactions were carried out using TaqMan Fast Advanced Master Mix (Applied Biosystems) per manufacturer's protocols, with a QuantStudio 5 Real-Time PCR System (Applied Biosystems). For mRNA, TaqMan Gene Expression Assays (Applied Biosystems) were used. For miRNA, TaqMan MicroRNA Assays (Applied Biosystems) were used. Amplification was performed with 1 cycle of 95 °C for 10 min and 50 cycles of 95 °C for 15 s and 60 °C for 60 s. All experiments were done in triplicate. For mRNA, relative quantification was done by normalizing to intrinsic GAPDH expression. For miRNA, relative quantification was carried out by normalizing to total RNA amount.

## Glioma-intrinsic classification of GBM patient tumors

To determine molecular subtypes and validate the selected RNA markers, we performed RNA sequencing analysis in patient tissue samples. Briefly, 1 µg of total RNA extracted from snap-frozen patient tumor tissue was used for RNA-seq library preparation using Illumina TruSeq Stranded Total RNA Library Prep Gold Kit according to the manufacturer's instructions. Library fragment size was determined by DNA 1000 Bioanalyzer (Agilent) and the libraries were quantified by qPCR using KAPA Library Quantification Kit (KAPA Biosystems). The libraries were pooled in equimolar concentration and cluster generation was performed on Illumina cBot system. Sequencing (150 bp paired-end) was performed by Illumina HiSeq 3000 system according to manufacturer's protocol. RNA sequencing reads were aligned with STAR v2.7.9a[51] to GRCh38 human reference genome. Gene expression levels measured in transcripts per kilobase million (TPM), were then calculated using RSEM v1.3.3[52]. Log2 transformation was further

applied to the TPM values for subsequent molecular classification. The molecular subtypes of GBM tumors were interrogated and determined with single-sample Gene Set Enrichment Analysis (ssGSEA) using glioma-intrinsic (GI) signatures as described by Wang et al. [7]. Briefly, for each sample, gene expression levels were rank-normalized and rank-ordered. Subtype was determined using the empirical $P$ values for the raw ssGSEA scores.

### Feature testing of combined gene signature

To evaluate the ability of the selected mRNA targets to classify the GI subtypes, the 22 EZ-READ genes and the 150 GI signature genes, with a common gene set of *LOX*, *TGFBI*, *PLAUR*, *COL1A2* and *CYP1B1*, were combined for further verification in a separate clinical study. A 5-fold cross-validation analysis, using pamr package v1.56.1[53], was performed on the gene expression data determined by tumor RNA sequencing to compute the threshold at which the misclassification error would be minimum. A threshold of 0.7 was determined to eliminate genes that do not contribute to the variability between the classifications, returning 144 out of 167 genes that are able to correctly predict the GI subtypes of samples. All EZ-READ targets except *ARC* contributed to the variability between classifications. Hierarchical clustering of classical ($n = 6$), mesenchymal ($n = 10$) and proneural ($n = 6$) samples with the 144 cross-validation threshold-filtered genes was carried out.

### Development and validation of EZ-READ plasma gene signature

To identify plasma gene signatures for GBM diagnosis and subtyping, employing the EZ-READ platform, we measured mRNA and miRNA expression levels of clinical plasma samples, and developed a multi-step linear regression scoring model to classify the disease status (Disease = 1, Control = 0) and mesenchymal subtype status (Mesenchymal = 1, Proneural/Classical = 0). For both classifications, the model was developed using a training patient cohort, where the markers were selected based on their classification performance (i.e., area under the curve (AUC) in receiver operating characteristic (ROC) analysis). As a comparison, we also developed a single-step multinomial regression model using the same set of markers identified through the multi-step model. We further evaluated the performance of both models (multi-step and single-step regression models) in an independent validation patient cohort and demonstrated that the multi-step model outperforms the single-step model.

### Gold standard classification of IDH1 mutation status in patient tumors

To determine the IDH1 mutation status, we performed tissue immunohistochemistry (IHC) staining and independently verified with tissue DNA sequencing. IDH1 mutation status was defined according to IDH1 R132H immunopositivity and mutation(s) in *IDH1* genes. For IHC, paraffin-embedded tumor sections were subjected to antigen retrieval with citrate buffer for 12 min then stained using the monoclonal antibody IDH1 R132H (Master Diagnostica, clone H09, 1:1000 dilution)[54]. For sequencing, genomic DNA was extracted from flash frozen primary tumors using the DNeasy Blood & Tissue Kit (Qiagen) in accordance with the manufacturer's protocol. Mutational alterations of *IDH1* at hotspot codons R132 were sequenced using the Sanger method after standard PCR amplification, as previously described[6]. Briefly, PCR products were purified using Wizard SV Gel and PCR Clean-up System (Promega, USA) and sequenced using the BigDye Terminator v3.1 Cycle Sequencing Kit (Applied Biosystems, USA). The primers used are as follows: *IDH1* Forward 5′-AATGAGCTCTATATGCCATCACTG-3′; *IDH1* Reverse 5′-TTCATACCTTGCTTAATGGGTGT-3′; *IDH1* sequencing 5′-AATGAGCTCTATATGCCATCACTG-3′.

### EZ-READ analysis of IDH1 mutation status

To assess the ability of the EZ-READ platform to determine patient IDH1 mutation status, we profiled tissue and paired plasma samples from IDH1 wild type and IDH1 R132H patients. Gold standard patient classification was performed through tissue IHC and verified by tissue DNA sequencing, as described above. Employing the EZ-READ transducers (Supplementary Table 2), we measured the expression levels of *IDH1* wild type and *IDH1 R132H* mRNA. Samples identified with the wild type but not *R132H* mRNA were classified as wild type; samples identified with both wild type and *R132H* mRNA were classified as mutant.

### Western blotting

Extracellular vesicles isolated by ultracentrifugation were lysed in radio-immunoprecipitation assay (RIPA) buffer containing protease inhibitors (Thermo Scientific) and quantified using bicinchoninic acid assay (BCA assay, Thermo Scientific). Protein lysates were resolved by sodium dodecyl sulfate polyacrylamide gel electrophoresis (SDS-PAGE), transferred onto polyvinylidene fluoride membrane (PVDF, Invitrogen), and immunoblotted with antibodies against protein markers: HSP90 (Cell Signaling, 1:1000 dilution), Flotillin 1 (BD Biosciences, 1:1000 dilution), CD63 (Santa Cruz, 1:250 dilution), ALIX (Cell Signaling, 1:1000 dilution), TSG101 (BD Biosciences, 1:1000 dilution) and LAMP-1 (BD Biosciences, 1:1000 dilution). Following incubation with anti-mouse IgG, HRP-linked antibody (Cell Signaling, 1:2000 dilution) or anti-rabbit IgG, HRP-linked antibody (Cell Signaling, 1:2000 dilution), as appropriate, enhanced chemiluminescence was used for immunodetection (Thermo Scientific).

### Hematoxylin and eosin tissue staining

Paraffin-embedded tissue sections of 5 μm thickness were deparaffinized, rehydrated and subsequently stained with hematoxylin. Following rinsing in tap water, the tissue section was stained with eosin and finally dehydrated and mounted with DPX mounting medium. Micrographs of stained tissue sections were acquired using Nikon Eclipse TE2000-S microscope.

### Electron microscopies

For scanning electron microscopy, samples were fixed with half-strength Karnovsky's fixative and washed twice with PBS. After dehydration in a series of increasing ethanol concentrations, samples were transferred for critical drying (Leica) and subsequently sputter-coated with gold (Leica), before imaging with a scanning electron microscope (JEOL 6701). For transmission electron microscopy, dried samples were imaged with a transmission electron microscope (JEOL 2200FS).

### Statistics & reproducibility

All measurements were performed in triplicate, and the data displayed as means ± standard deviation, unless otherwise stated. Significance tests were carried out with a two-tailed Student's $t$ test. For inter-sample comparisons, multiple pairs of samples were tested, and the resulting $P$ values were adjusted for multiple hypothesis testing using Bonferroni correction. Values that had an adjusted $P < 0.05$ were determined as significant. Correlation analysis was performed with linear regression to determine the goodness-of-fit ($R^2$), unless otherwise stated. Statistical analyses were performed using R (v.4.0.3) and GraphPad Prism (v.9.0.0). For clinical analysis of GBM diagnosis and subtyping, sample sizes were restricted by the availability of patient samples. No data were excluded from the analyses. Plasma samples were randomized into two representative cohorts. All experiments were performed blinded from the clinical diagnoses.

### Reporting summary

Further information on research design is available in the Nature Portfolio Reporting Summary linked to this article.

## Data availability

The data supporting the results in this study are available within the paper and its Supplementary Information. Source data are provided

with this paper. To protect patient privacy, the Ministry of Health (MOH) of Singapore has imposed regulations on all healthcare institutions that no raw patient sequencing data can be stationed outside of the hospital network due to potential risks of re-identifying patients. Informed patient consent was obtained in accordance with these regulations, meaning that consent to data deposition in a public repository was not provided. Raw sequencing data that support the findings of this study are therefore available upon request due to patient privacy protection. Request for data access should be directed to Beng Ti Ang (ang.beng.ti@singhealth.com.sg). Review of the request will be completed within two months. Upon request approval by the National Neuroscience Institute Tissue Bank, data will be accessible for research use only. The processed RNA sequencing data can be downloaded from Synapse under the project ID: syn51691297. Source data are provided with this paper.

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

## Acknowledgements

The authors thank clinical team members from National Neuroscience Institute, Singapore, for patient sample collection, and Olivia Seow for critically proofreading the manuscript. This work was supported in part by funding from National University of Singapore (NUS), NUS Research Scholarship, Ministry of Education, Institute for Health Innovation & Technology, and National Medical Research Council under both the Translational and Clinical Research (TCR) Flagship Programme – Tier 1 (NMRC/TCR/016-NNI/2016) and the Open Fund-Large Collaborative Grant (OF-LCG) – Tier 1 (MOH-000541-00) awarded to B.T.A.

## Author contributions

Y.Z., C.Y.W., C.Z.J.L. and H.S. designed the study, performed data analysis and wrote the manuscript. Y.Z., C.Y.W., C.Z.J.L., Q.C., Z.Y., A.N. and Z.W. performed the research. Q.Y.P. performed bioinformatics analysis on sequencing data. S.W.L. performed histological analyses. T.P.L., B.T.A. and C.T. provided the de-identified clinical samples and health information. B.T.A. and C.T. wrote the manuscript and established the banked resource of de-identified patient GBM tumors and associated biofluids. They also established the glioma-propagating cells, acquired bulk RNA sequencing for all samples and verified the initial glioma-intrinsic subtype signature. All the authors contributed to the manuscript.

## Competing interests

The authors declare no competing interests.
