## [Peer Review File · Nature Communications]

Multiplexed RNA profiling by regenerative catalysis enables blood-based subtyping of brain tumorsREVIEWER COMMENTS

Reviewer #1 (Remarks to the Author):

The manuscript is technically strong and the authors propose a new technique for RNA marker analysis. Of particular interest is the minimal nature of sample preparation in the technique.

However, there are some weaknesses on the clinical validation aspect of the manuscript.

The 'Mesenchymal' and 'Proneural/Classical' subtypes are generally not used clinically. A much more important classification in glioma relates to IDH-1 status. The authors be familiar with the WHO2016 and WHO2021 guidelines that highlight the fact that IDH status is a major bifurcation in the classification of gliomas, with major prognostic and therapeutic implications. Furthermore, the IDH subclass has a major bifurcation depending on TERT mutation status into the astrocytic and oligodendroglial subclasses. The authors could address this by assessing whether their technology could accurately allocate patients into these important clinical groups and reporting on these key molecular subgroups in their analysis. At the very least, the manuscript should report on these key mutations (IDH/TERT).

The authors do not specifically comment on the rigor and reproducibility of their analysis. Is the analysis reproducible by multiple individual laboratory members with the same results. Has this technique been tried on additional patient samples besides a single validation cohort? Measurements are reported in triplicate, but were they repeated by independent practitioners?

Reviewer #2 (Remarks to the Author):

In this manuscript, Wang and colleagues developed an extracellular vesicle (EV) analysis platform (termed EZ-READ) based on nanoparticles and microfluidics to diagnose and even subtype glioblastoma by digital quantification of EV RNA in blood specimens. HRP-encapsulated ZIF-8 was linked to magnetic beads via DNA probes, constituting a sensitive biosensor for RNA. The RNA signal is converted to the HRP@ZIF-8 nanoparticle signal by selective cleavage of the RNA-bound DNA single strand by duplex-specific nuclease (DSN). In this process, the HRP@ZIF-8 nanoparticles work in a target-recyclable manner, acting as a catalyst to achieve detection signal amplification. On this basis, the author introduced a microfluidic chip and used the principle of digital PCR to further improve the detection sensitivity. Compared with the commonly used droplet microfluidics, this microwell structure will have more advantages in terms of portability of operation and control of parameters. Using this platform, the authors profiled 22 mRNA markers and 10 miRNA markers derived from EVs from 6 GBM cell lines. Compared with clinical blood samples and tissue samples, 6 highly relevant RNA markers were selected for clinical validation. In a sample of 23 GBM patients and 16 healthy controls, the feasibility of the platform was demonstrated, including tumor diagnosis and subtype classification.

This platform avoids the cumbersome steps of traditional RNA detection (such as PCR), and uses a clever signal amplification strategy and digital microfluidic mode to achieve high-sensitivity detection of RNA, which has important inspiration for the nucleic acid detection community. The overall design of the platform is innovative and the capacity of the developed platform is extremely useful for GBM diagnosis and subtype classification. While the authors did an excellent job in demonstrating the detection strategy of the HRP@ZIF-8 complex, the part of the manuscript addressing the microfluidic manipulation is notably less convincing. The paper is concise, and includes a wealth of constructive criticism that is worth sharing. Prior to publication, I recommend with the following comments.

1. The manuscript lacks a description of the overall structure of the microfluidic chip and the number of microwells. Considering the authors' approach to utilizing digital microfluidics, both the number of microwells and the chip's footprint are critical factors. Since the flow resistance of each microwell is different, how to ensure that each microwell is evenly loaded with samples when adding solution. It is recommended to add large-scale microscopic images in SI to confirm the feasibility of the protocol.
2. This was an operational challenge considering that the authors used a vacuum to load the samples. How to ensure that the HRP@ZIF-8 solution does not fulfill the microwells, but the substrate buffer just fills the microwells. Is air sealing achieved by further evacuation? Considering the high gas permeability of PDMS, liquid evaporation cannot be avoided during the detection process. During the detection process, how long can the solution be maintained in the microwells?
3. Different from the well-known dodecahedral shape of ZIF-8, the HRP@ZIF-8 synthesized by the authors seem to have poor monodispersity, as shown in Figure 1B (50-300 nm) and Figure S4 (~200 nm derived from TEM, and ~300 nm derived from DLS). It is recommended that the authors add the XRD simulation value of ZIF-8 to better demonstrate the crystallinity of the synthesized HRP@ZIF-8.

4. The authors stated that HRP@ZIF-8 showed stronger enzymatic activity than free HRP arising from enzyme conformational changes. This conclusion needs to be established under the same enzyme concentration. Considering that HRP@ZIF-8 needs to be synthesized, determining the enzyme concentration of HRP with ZIF-8 is a key issue.
5. The authors stated that raw images were processed in gray scale, and fluorescence signals were pseudo-colored red. It means that both Figure 1c and 3c are merged grayscale images, and that the pseudo-colored red or blue is manually added later? Is subsequent image processing, especially digital quantitation, manual?
6. As the authors declared that HRP@ZIF-8 showed improved enzyme activity and stability in complex environments than equal concentration of HRP, please describe the quantitative methods specifically.
7. In Supplementary Figure 4d, different concentrations of hydrogen peroxide, corresponding to lines of different colors, should be marked one by one.
8. In Supplementary Figure 5d, scale bar of the microfluidic chip was lost.
9. The authors tested the correlation of fluorescence signal in the microfluidic chip and different numbers of HRP@ZIF-8 nanoparticles in Supplementary Figure 5d and performed in triplicate, it is recommended to show the results in the form of intuitive charts and the analytical performance for quantifying HRP@ZIF-8 nanoparticles in microfluidic chip (such as, LOD and maximum nanoparticles number identification limit) should be included furtherly.
10. The authors declared that the amplification efficiency was 93.4%, please describe the quantitative methods specifically.

Reviewer #3 (Remarks to the Author):

The authors present a novel approach for detecting glioblastoma through disease-specific mRNA and miRNA levels in blood plasma. This well-designed study incorporates robust controls and impressive results. Specifically, the authors develop a transducer based on HRP encapsulated in a MOF that is cleaved from a magnetic bead in the presence of target RNA. The resulting HRP-encapsulated nanoparticle catalyzes chemifluorescence of QuantaRed in a custom-designed microfluidic device, with the fluorescence signal quantified using a microscope. The authors utilize RT-qPCR to measure mRNA

and miRNA levels and screen for biomarkers for glioblastoma, including subtypes. The method is clinically validated using 39 real samples. The authors are to be commended for their detailed and well-executed study. However, I have a few technical questions, comments, and suggestions that could further improve the quality of the manuscript:

1. The most critical question I have is about the stage of cancer the patients were when the samples were extracted. It would be beneficial to regroup patients based on the stages of cancer to perform validation. Knowing the sensitivity of this test across disease prognosis and how much it can aid in early-stage detection would be valuable information.
2. What is the data size (total number of vesicles) shown in histograms in Figure 7b? This information should be included in the figure.
3. How many base pairs do mRNA and miRNA make with the DNA linker, and how did the authors confirmed target recycling? Target recycling requires optimization of the sequence and length so that, upon cleaving, the RNA dissociates from both DNA strands. Have the authors optimized the sequence, or have they conducted any other assays to ensure target recycling is occurring?
4. Regarding the previous question, in Supplementary Figure 1c, the authors should provide evidence that the increase in signal over time is not due to the slow rate of reaction.
5. A workflow illustration/schematics for the clinical sample would be appreciated.
6. Is absolute quantification of RNA levels in human samples possible from the digital counting, perhaps using a calibration curve?
7. The authors should include a comparison of the sensitivity of their technique to similar systems for microRNA detection based on FRET.
8. A paragraph discussing the outlook of this study, including the required steps to obtain regulatory approval for clinical testing and the challenges associated with commercializing this method, should be included in the discussion.
9. The microfluidic chamber used in this study is impressive. It is a new design, and the authors should emphasize its advantages more clearly in the main text.
10. A short video of sample loading and video from microscope of the sample fluorescence change over time would be a nice addition.

Other minor comments:

1. The authors should consider analyzing their use of the word "faithful" in the manuscript. "Reliable" may be a more appropriate term, as faithful suggests loyalty and is likely unsuitable for describing non-living objects or methods.
2. The negative aspects of PCR are overstated in the introduction. For example, things like "exquisite sequence design" are not issues, as they only have to be done once for a particular RNA.

REVIEWER COMMENTS

Reviewer #1 (Remarks to the Author):

The manuscript is technically strong and the authors propose a new technique for RNA marker analysis. Of particular interest is the minimal nature of sample preparation in the technique.

However, there are some weaknesses on the clinical validation aspect of the manuscript.

The 'Mesenchymal' and 'Proneural/Classical' subtypes are generally not used clinically. A much more important classification in glioma relates to IDH-1 status. The authors be familiar with the WHO2016 and WHO2021 guidelines that highlight the fact that IDH status is a major bifurcation in the classification of gliomas, with major prognostic and therapeutic implications. Furthermore, the IDH subclass has a major bifurcation depending on TERT mutation status into the astrocytic and oligodendroglial subclasses. The authors could address this by assessing whether their technology could accurately allocate patients into these important clinical groups and reporting on these key molecular subgroups in their analysis. At the very least, the manuscript should report on these key mutations (IDH/TERT).

We thank the reviewer for the valuable feedback and provide further clarification and experimental validation as recommended. Indeed, as the reviewer pointed out, IDH status is a major bifurcation in the WHO2021 classification of glioma tumors, with IDH-wild type defining primary adult glioblastoma (GBM), and IDH mutations enriched in oligodendroglioma (grades 2, 3) and astrocytoma (grades 2, 3, 4) tumors and are typically associated with better prognosis [Louis, D.N. et al. *Neuro-oncology*, 23(8), 1231-1251 (2021)]. Our present study focuses on primary adult GBM, where morphologically identical patient tumors demonstrate extensive molecular heterogeneity based on transcriptomic profiling. Indeed, through RNA profiling of primary tumors, recent consortial efforts have supported the importance of the glioma-intrinsic (GI) transcriptomic subtypes (i.e., mesenchymal, proneural and classical) in GBM and its potential role in accounting for inter-patient variability in prognosis and treatment response [The Cancer Genome Atlas Research Network. *Nature*, 455, 1061-1068 (2008)] [Varn, F.S. et al. *Cell*, 185(12), 2184-2199 (2022)]. Specifically, patients who demonstrated subtype switching from non-mesenchymal (i.e., proneural and classical) to mesenchymal profile showed poorer survival compared to patients without the mesenchymal subtype switching [Wang, Q. et al. *Cancer Cell*, 32(1), 42-56 (2018)]. Furthermore, recent multi-omic studies have also identified therapeutic vulnerabilities correlated with the GI transcriptomic subtypes [Migliozzi, S. et al. *Nat. Cancer*, 4, 181-202 (2023)]. Motivated by these studies on molecular heterogeneity of GBM subtypes, noting that they were performed solely through tumor tissue analysis, we applied the EZ-READ platform for multiplexed profiling of circulating RNA markers, reasoning that the technology's direct detection could facilitate minimally-invasive molecular characterization of the disease.

We also thank the reviewer for your recommendation on IDH mutation classification. We believe this experimental validation has further strengthened the manuscript. As suggested, we employed the EZ-READ platform to profile *IDH1* wild type and *R132H* mRNA in additional patient specimens with known IDH status ($n = 10$ patients, 7 with *IDH1* wild type and 3 with *IDH1 R132H* mutation) (Fig. R1, also new Supplementary Fig. 10c). The IDH subclass was clinically determined by standard immunohistochemistry and verified by DNA sequencing of the tumor tissues; these results were applied as the clinical gold-standard for tumor classification. For EZ-READ

Fig. R1. EZ-READ analysis of IDH1 mutation status. Tumor tissue and paired plasma samples ($n = 10$ patients, 7 with *IDH1* wild type and 3 with *IDH1 R132H* mutation) were analyzed through the EZ-READ platform. Gold-standard patient classification was performed through tumor tissue immunohistochemistry and sequencing. EZ-READ analysis of tumor tissue and plasma achieved an overall accuracy of 100% and 90%, respectively.

analysis, we designed specific probes to detect the *IDH1* wild type and *R132H* mRNA, respectively. Samples identified with the wild type but not *R132H* mRNA were classified as wild type tumors; samples identified with both wild type and *R132H* mRNA were classified as mutant tumors. The EZ-READ probe sequences employed for this mutation study are included in the revised **Supplementary Table 2**. Employing the EZ-READ platform, we characterized *IDH1* mRNA (wild type and *R132H*, respectively) in patient tumor tissues ($n = 10$) as well as paired plasma samples ($n = 10$). The EZ-READ analysis of the tumor tissues corresponded accurately to the gold-standard classification (overall classification accuracy = 100%). For the plasma analysis, EZ-READ detected the wild type mRNA in all plasma samples and the *R132H* mRNA in 2/3 of the mutant plasma samples (overall classification accuracy = 90%). We further note the low concentration of *IDH1* mRNA in circulation (> 50 fold lower than other circulating mRNAs measured e.g., *LOX*, *SLCO1A3* and *TGFBI*). This finding is in agreement with published studies on the characterization requirement of *IDH1* mRNA in large volumes of blood and CSF samples [Chen, W.W. et al. *Mol. Ther. Nucleic*, 2, e109 (2013)][Reátegui, E. et al. *Nat. Commun.*, 9(1), 175 (2018)].

The authors do not specifically comment on the rigor and reproducibility of their analysis. Is the analysis reproducible by multiple individual laboratory members with the same results. Has this technique been tried on additional patient samples besides a single validation cohort? Measurements are reported in triplicate, but were they repeated by independent practitioners?

This is another excellent point. As recommended by the reviewer, we performed additional analytical and clinical validation experiments to assess the technology rigor and reproducibility. For analytical validation, we employed the EZ-READ platform to measure the same plasma samples (patient and control samples), by different users on different sensor chips (**Fig. R2**, also **new Supplementary Fig. 10b**). The results showed robust analytical performance with small coefficients of variation (within user = 2.04% and between users = 3.83%).

For additional clinical validation, we performed EZ-READ analysis on a new patient cohort ($n = 21$, 11 GBM and 10 control samples) (**Fig. R3**). The new clinical data support our initial findings and conclusions, that the EZ-READ platform could be applied to clinical plasma samples for diverse RNA profiling and that our previously identified EZ-READ signature performs consistently to distinguish and subtype GBM patients in the new clinical cohort (AUC for disease detection = 0.882, disease subtyping = 0.917). We have also included a combined clinical analysis ($n = 60$, 34 disease and 26 control samples) in **Fig. R4** (also **new Fig. 5e–f** and **new Supplementary Fig. 16**) to provide an overview of the clinical performance. Accordingly, we have updated the patient information (**new Supplementary Table 3**) and sample profiles (**new Supplementary Fig. 15**) to reflect the expanded validation cohort.

Fig. R2. Analytical evaluation of EZ-READ. The same samples were measured by different users on different EZ-READ chips. The technology showed robust performance with small coefficients of variation (within user = 2.04% and between users = 3.83%).

Fig. R3. Clinical validation of EZ-READ in additional patient cohort. A second validation cohort ($n = 21$ subjects: 11 disease (3 mesenchymal and 8 proneural/classical) and 10 control) was evaluated. (a) Disease classification. The markers *LOX*, *SLCO3A1* and *miR-17-5p* were measured with EZ-READ. The three-marker model generated from the training cohort was able to classify the disease and control samples. Receiver operating characteristic (ROC) curve analysis demonstrated good AUC. (b) Mesenchymal classification. The markers *miR-21-5p*, *miR-27a-3p* and *ALDH1A3* were measured with EZ-READ. The three-marker model generated from the training cohort was able to classify the mesenchymal samples. ROC curve analysis demonstrated good AUC. All measurements were performed in triplicate, and the data are displayed as mean \pm s.d. in a–b. ns = not significant, * $P < 0.05$ and *** $P < 0.0005$, Student's t test.

Fig. R4. Overview of clinical performance. (a–b) Receiver operating characteristic (ROC) curves of the Disease regression model (a) and the Mesenchymal regression model (b) on the training, expanded validation and combined patient cohorts, respectively. The expanded validation cohort consisted of $n = 40$ subjects: 22 disease (8 mesenchymal and 14 proneural/classical) and 18 control. (c) Comparison against single-step multinomial regression. Using the six derived markers (*miR-17-5p*, *miR-21-5p*, *miR-27a-3p*, *LOX*, *ALDH1A3* and *SLCO3A1*), the multi-step EZ-READ model was able to accurately classify the validation clinical samples into control, proneural/classical and mesenchymal samples (78% accuracy). The single-step multinomial regression model was unable to accurately classify the validation samples into the different groups (48% accuracy).

Reviewer #2 (Remarks to the Author):

In this manuscript, Wang and colleagues developed an extracellular vesicle (EV) analysis platform (termed EZ-READ) based on nanoparticles and microfluidics to diagnose and even subtype glioblastoma by digital quantification of EV RNA in blood specimens. HRP-encapsulated ZIF-8 was linked to magnetic beads via DNA probes, constituting a sensitive biosensor for RNA. The RNA signal is converted to the HRP@ZIF-8 nanoparticle signal by selective cleavage of the RNA-bound DNA single strand by duplex-specific nuclease (DSN). In this process, the HRP@ZIF-8 nanoparticles work in a target-recyclable manner, acting as a catalyst to achieve detection signal amplification. On this basis, the author introduced a microfluidic chip and used the principle of digital PCR to further improve the detection sensitivity. Compared with the commonly used droplet microfluidics, this microwell structure will have more advantages in terms of portability of operation and control of parameters. Using this platform, the authors profiled 22 mRNA markers and 10 miRNA markers derived from EVs from 6 GBM cell lines. Compared with clinical blood samples and tissue samples, 6 highly relevant RNA markers were selected for clinical validation. In a sample of 23 GBM patients and 16 healthy controls, the feasibility of the platform was demonstrated, including tumor diagnosis and subtype classification.

This platform avoids the cumbersome steps of traditional RNA detection (such as PCR), and uses a clever signal amplification strategy and digital microfluidic mode to achieve high-sensitivity detection of RNA, which has important inspiration for the nucleic acid detection community. The overall design of the platform is innovative and the capacity of the developed platform is extremely useful for GBM diagnosis and subtype classification. While the authors did an excellent job in demonstrating the detection strategy of the HRP@ZIF-8 complex, the part of the manuscript addressing the microfluidic manipulation is notably less convincing. The paper is concise, and includes a wealth of constructive criticism that is worth sharing. Prior to publication, I recommend with the following comments.

1. The manuscript lacks a description of the overall structure of the microfluidic chip and the number of microwells. Considering the authors' approach to utilizing digital microfluidics, both the number of microwells and the chip's footprint are critical factors. Since the flow resistance of each microwell is different, how to ensure that each microwell is evenly loaded with samples when adding solution. It is recommended to add large-scale microscopic images in SI to confirm the feasibility of the protocol.

We thank the reviewer for the valuable feedback and have revised the manuscript accordingly to clarify the design of the microfluidic chip. To ensure that the microchannels have the same length and fluidic resistance, the microfluidic chip features a fractal branching design. Specifically, the fluidic channel is repeatedly divided and subdivided into thousands of branches. Each branch is terminally connected to four microwells (**Fig. R5a**, also **new Supplementary Fig. 6c**). This design thus ensures equal flow resistance to individual microwells, facilitates even microwell loading (e.g., with HRP@ZIF-8 particles and reagents) and enables scalable expansion of the number of microwells. With six layers of fractal branching, the current prototype houses a total of 4096 microwells in a 2.5 cm x 2.5 cm footprint; additional branching can be included to increase the number of microwells. As recommended, we have also included a large-scale microscopy image of the microfluidic device. Magnified views over different spatial regions of the device demonstrate uniform reagent (red dye) loading and even distribution in individual microwells (**Fig. R5b**, also **new Supplementary Fig. 6d**).

Fig. R5. Fractal branching microfluidic design. (a) Schematic of the microfluidic design. The chip features six layers of fractal branching, where the main fluidic channel is repeatedly divided and subdivided into thousands of branches. Each branch is terminally connected to four microwells. This design thus achieves equal fluidic length and resistance to individual microwells, facilitates even microwell loading, and enables scalable expansion. (b) Large-scale device image. Magnified views over different spatial regions demonstrate uniform reagent loading and even distribution in individual microwells.

2. This was an operational challenge considering that the authors used a vacuum to load the samples. How to ensure that the HRP@ZIF-8 solution does not fulfill the microwells, but the substrate buffer just fills the microwells. Is air sealing achieved by further evacuation? Considering the high gas permeability of PDMS, liquid evaporation cannot be avoided during the detection process. During the detection process, how long can the solution be maintained in the microwells?

This is an excellent point and we further clarify the operational optimization. To ensure uniform microwell filling with HRP@ZIF-8 solution and subsequent loading with substrate solution (i.e., to prevent complete filling by HRP@ZIF-8 solution), we first conducted optimization experiments using color dye solutions to characterize the volume occupancy in channels and microwells with respect to fluid introduction at the inlet. Based on the characterization data, we found that an introduction of 5 μl of HRP@ZIF-8 solution at the inlet for 2.5 min, followed by 20 μl of substrate buffer for 5.5 min, achieved an optimal loading of both solutions into the microwells. Finally, for air sealing, as the PDMS chamber remained negatively pressured after the substrate loading, we introduced air flow at the inlet to accomplish complete microwell sealing.

Fig. R6. Liquid occupancy in microwells. (a) Schematic of the microfluidic device. The PDMS chamber was covered with an impermeable, optical adhesive film to reduce liquid evaporation. (b) Experimental characterization. We monitored the liquid occupancy in the microwells. High occupancy can be maintained for at least several hours, which is significantly longer than the assay duration (~30 min). All measurements were performed in triplicate, and the data are displayed as mean \pm s.d.

With respect to PDMS gas permeability, we provide further clarification and experimental evaluation. As illustrated in the device schematic (Fig. R6a, also new Supplementary Fig. 7a), the PDMS chamber was covered with a MicroAmp optical adhesive film to reduce liquid evaporation; this impermeable film is typically used to seal PCR reactions during high-temperature thermal cycling. We punctured the film only at the fluidic inlet to introduce reagents; the rest of the PDMS chamber remained covered to maintain negative pressure and reduce evaporation. We also performed new experimental characterization to evaluate the degree of liquid evaporation from the filmed microfluidic platform (Fig. R6b, also new Supplementary Fig. 7b). We found that solution can be maintained within the microwells for at least several hours at room temperature. As the entire EZ-READ assay can be completed much faster (~30 min), the microwells experience minimal volume changes meanwhile.

3. Different from the well-known dodecahedral shape of ZIF-8, the HRP@ZIF-8 synthesized by the authors seem to have poor monodispersity, as shown in Figure 1B (50-300 nm) and Figure S4 (~200 nm derived from TEM, and ~300 nm derived from DLS). It is recommended that the authors add the XRD simulation value of ZIF-8 to better demonstrate the crystallinity of the synthesized HRP@ZIF-8.

We thank the reviewer for this helpful comment. As suggested, we performed XRD simulation of pure ZIF-8. In addition, we experimentally measured the XRD spectra of ZIF-8 as well as HRP@ZIF-8. Our experimental data matched well with the simulation result, confirming the crystallinity of the synthesized HRP@ZIF-8 (Fig. R7, also new Supplementary Fig. 4b). We also note the morphology differences between ZIF-8 and HRP@ZIF-8. This can be attributed to the differential kinetics of ZIF-8 framework assembly mediated by HRP incorporation, thereby promoting the formation of more spherical products in HRP@ZIF-8. We and others have previously reported on this phenomenon and the resultant morphology changes [Wang, Z. et al. *Nat. Commun.*, 12(1), 4039 (2021)][Wang, L. et al. *ACS Sustain. Chem. Eng.*, 7(17), 14611-14620 (2019)].

4. The authors stated that HRP@ZIF-8 showed stronger enzymatic activity than free HRP arising from enzyme conformational changes. This conclusion needs to be established under the same enzyme concentration. Considering that HRP@ZIF-8 needs to be synthesized, determining the enzyme concentration of HRP with ZIF-8 is a key issue.

This is another excellent point. We provide further experimental clarification on our determination of enzyme concentration. Specifically, to ensure equivalent enzyme concentrations were compared across experiments, we used fluorescently-labeled HRP; FITC-labeled HRP was used due to its distinctive fluorescence profile from the final chemifluorescent product. Enzyme concentration was determined by measuring the FITC fluorescence intensity in pristine HRP solution as well as in HRP@ZIF-8. In particular, to reduce potential inaccuracy in fluorescence measurement that could arise from ZIF-8 encapsulation, we dissolved the formed structures to release the embedded HRP before fluorescence characterization.

Fig. R7. Simulation and experimental XRD. In comparison to the simulated XRD spectrum of ZIF-8, our experimental measurements of ZIF-8 and HRP@ZIF-8 showed corresponding XRD peaks, confirming the crystallinity of the prepared HRP@ZIF-8.

5. The authors stated that raw images were processed in gray scale, and fluorescence signals were pseudo-colored red. It means that both Figure 1c and 3c are merged grayscale images, and that the pseudo-colored red or blue is manually added later? Is subsequent image processing, especially digital quantitation, manual?

We thank the reviewer for this comment and provide further clarification. We used a Leica DFC9000 monochrome microscopy with CMOS camera for fluorescence imaging. The fluorescence images were thus captured in greyscale and automatically processed (pseudo-colored red or blue) using the LAS X software according to the greyscale intensity. The processed images were then merged with bright field images for colocalization analysis in **Fig. 3c**. For digital quantitation, the greyscale images were used directly for ImageJ analysis (particle counting function) to acquire a distribution of fluorescence intensity in individual wells. We used this automated analysis to establish threshold and determine the number of positive wells. We have clarified the approach in the revised manuscript.

6. As the authors declared that HRP@ZIF-8 showed improved enzyme activity and stability in complex environments than equal concentration of HRP, please describe the quantitative methods specifically.

We thank the reviewer for this helpful comment and provide further experimental details. First, to compare the enzyme activity in different formulations (e.g., pristine HRP vs. HRP@ZIF-8) and in different environments (e.g., PBS vs. DSN buffer), we ensured equivalent enzyme concentrations were applied and evaluated. As discussed in our response to comment 4, we used fluorescently-labeled HRP to quantify enzyme concentration and ensure concentration-matched comparisons. Second, to evaluate catalytic activity in these comparisons, we used a commercial kit (QuantaRed Enhanced Chemifluorescent HRP Substrate, Thermo Scientific) to assess the HRP activity. Specifically, HRP catalysis produces a distinct chemifluorescence signal, through the generation of esorufin, a soluble fluorescent product with excitation/emission maxima at 570/585 nm. HRP enzyme activity was determined by measuring the emission intensity after incubating the substrate with enzyme formulations. We employed both end-point measurements (at 5 min) as well as kinetic measurements (continuous fluorescence measurements for 20 min) to evaluate product formation. For example, to evaluate catalytic performance in different environments (e.g., PBS vs. DSN buffer), we prepared concentration-matched formulations (pristine HRP vs. HRP@ZIF-8) in different buffer conditions and incubated the preparations with equivalent amounts of chemifluorescent HRP substrate. We performed kinetic measurements (continuous fluorescence measurements for 20 min) to evaluate reaction activity. At the end of the measurement, we evaluated the hydrodynamic diameter of HRP@ZIF-8 to confirm there was no particle disintegration after the incubation.

7. In Supplementary Figure 4d, different concentrations of hydrogen peroxide, corresponding to lines of different colors, should be marked one by one.

As suggested, we have indicated the specific concentrations of hydrogen peroxide in the revised **Supplementary Fig. 4d**.

8. In Supplementary Figure 5d, scale bar of the microfluidic chip was lost.

As suggested, we have included the scale bar accordingly in the revised **Supplementary Fig. 7d**.

9. The authors tested the correlation of fluorescence signal in the microfluidic chip and different numbers of HRP@ZIF-8 nanoparticles in Supplementary Figure 5d and performed in triplicate, it is recommended to show the results in the form of intuitive charts and the analytical performance for quantifying HRP@ZIF-8 nanoparticles in microfluidic chip (such as, LOD and maximum nanoparticles number identification limit) should be included furtherly.

This is another excellent suggestion. Accordingly, we provide more intuitive charts to illustrate the analytical performance data for quantifying HRP@ZIF-8 nanoparticles (**Fig. R8**, and a **new Supplementary Fig. 8**). We further include the analytical characteristics. The limit of detection was determined ~10 particles, as defined by 3x the standard deviation of a matched control (no particle control). Likewise, the maximum detection limit was determined ~4000 particles, based on the dynamic distinguishable signal range. We have clarified these in the revised manuscript.

Fig. R8. Quantifying HRP@ZIF-8 nanoparticles. The limit of detection (LOD, dotted line) was determined ~10 particles, as defined by 3x the standard deviation of a matched control (no particle control). The maximum detection limit was estimated ~4000 particles based on the dynamic distinguishable signal range. All measurements were performed in triplicate, and the data are displayed as mean \pm s.d.

10. The authors declared that the amplification efficiency was 93.4%, please describe the quantitative methods specifically.

We provide further experimental clarification. The EZ-READ platform employs DSN to recognize RNA-DNA heteroduplexes and cleave the DNA linkers to release the attached HRP@ZIF-8 nanoparticles for RNA quantification. As a gold standard to determine maximal DSN cleavage efficiency, we used free-floating fluorescent DNA probes to form the RNA-DNA heteroduplexes and treated the mixture with DSN. We measured the amount of cleaved fluorescent probes to determine the maximal amplification efficiency (100%). In comparison, to characterize the EZ-READ platform, equal concentration of magnetic bead-immobilized DNA probes were applied for DSN cleavage. The EZ-READ amplification efficiency was normalized to that of the free-floating gold standard and measured 93.4%. The reduced efficiency is likely due to decreased enzyme accessibility due to probe immobilization on the beads.

Reviewer #3 (Remarks to the Author):

The authors present a novel approach for detecting glioblastoma through disease-specific mRNA and miRNA levels in blood plasma. This well-designed study incorporates robust controls and impressive results. Specifically, the authors develop a transducer based on HRP encapsulated in a MOF that is cleaved from a magnetic bead in the presence of target RNA. The resulting HRP-encapsulated nanoparticle catalyzes chemifluorescence of QuantaRed in a custom-designed microfluidic device, with the fluorescence signal quantified using a microscope. The authors utilize RT-qPCR to measure mRNA and miRNA levels and screen for biomarkers for glioblastoma, including subtypes. The method is clinically validated using 39 real samples. The authors are to be commended for their detailed and well-executed study. However, I have a few technical questions, comments, and suggestions that could further improve the quality of the manuscript:

1. The most critical question I have is about the stage of cancer the patients were when the samples were extracted. It would be beneficial to regroup patients based on the stages of cancer to perform validation. Knowing the sensitivity of this test across disease prognosis and how much it can aid in early-stage detection would be valuable information.

We thank the reviewer for the insightful comments. In the current study, we focus on molecular characterization of glioblastoma (GBM), which is a Grade 4 glioma. Recent studies show that molecular subtyping of GBM provides new clinical opportunities for patient stratification (e.g., prognosis and treatment response) [The Cancer Genome Atlas Research Network. *Nature*, 455, 1061-1068 (2008)] [Varn, F.S. et al. *Cell*, 185(12), 2184-2199 (2022)]. Noting that these studies were performed solely through tumor tissue analysis, we thus applied the EZ-READ platform for multiplexed profiling of circulating RNA markers, reasoning that the technology's direct detection could facilitate minimally-invasive molecular characterization of the disease. The reviewer brought up an excellent point for early-stage cancer detection. We evaluate this potential capability based on our current analytical and clinical performance. The EZ-READ platform is a direct, sensitive and versatile technology (limit of detection ~9 RNA copies, comparable to qPCR performance but bypasses extensive sample processing; and is widely applicable for different types of circulating RNA targets including miRNA, mRNA and lncRNA). As supported by published studies that leverage circulating RNA for early stage cancer detection (e.g., TisN0M0 colon cancer [Min, L. et al. *J. Extracell. Vesicles*, 8(1), 1643670 (2019)] and stage 1 non-small cell lung cancer [Jin, X. et al. *Clin. Cancer Res.*, 23(17), 5311-5319 (2017)][Guo, W. et al. *Transl. Lung Cancer Res.*, 11(4), 572 (2022)]), the EZ-READ features good analytical performance to match to biomarker concentration and broad target compatibility for early-stage cancer detection.

2. What is the data size (total number of vesicles) shown in histograms in Figure 7b? This information should be included in the figure.

As suggested, we have included the total number of EVs measured in the associated figure caption. Specifically, nanoparticle tracking analysis (NTA) system was used for EV quantification. All measurements were done with identical system settings for consistency. Individual EV samples were diluted to obtain ~50 vesicles in the field of view to achieve optimal counting and recording, resulting in ~10⁹ particles being evaluated in the histogram analysis.

3. How many base pairs do mRNA and miRNA make with the DNA linker, and how did the authors confirmed target recycling? Target recycling requires optimization of the sequence and length so that, upon cleaving, the RNA dissociates from both DNA strands. Have the authors optimized the sequence, or have they conducted any other assays to ensure target recycling is occurring?

This is another excellent point. We provide further schematic clarification and experimental evidence to highlight the advantage of the EZ-READ DSN system (as compared to sequence-specific DNA cleavage) in producing short and easily dissociated DNA cuts. Specifically, in the EZ-READ platform, mRNA and miRNA target form 20-23 base pairs with the DNA linker; DSN recognizes the RNA-DNA heteroduplex and cleaves the DNA strand in a random and repeated

fashion to yield short DNA products ≤ 6 nt [Shagin, D.A. et al. *Genome Res.*, 12(12), 1935-1942 (2002)] (**Fig. R9**, left, also new **Supplementary Fig. 5a**). As these short DNA products have very low melting temperatures, they readily dissociate from the RNA target to facilitate target cycling. The DSN cleavage mechanism presents distinct advantages over sequence-specific DNA cleavage. In sequence-specific DNA cleavage (e.g., BstNI, with cut site CC/WGG), the enzyme recognizes specific cut sites of the RNA-DNA heteroduplex to cleave the DNA into distinct products; the process thus requires much more extensive sequence selection to facilitate the optimization of the DNA product length and dissociation of the products. To experimentally validate the mechanism differences, we incubated identical RNA-DNA heteroduplexes (**new Supplementary Table 1**) with DSN or BstNI, extracted and characterized the formed DNA products with polyacrylamide gel electrophoresis (PAGE) analysis (**Fig. R9**, right, also new **Supplementary Fig. 5a**). In the DSN treatment, we observed no distinct DNA products, indicating random repeated cleavage of the 23 nt DNA probe. Note that DNA products ≤ 6 nt are too short for PAGE characterization. In contrast, the BstNI sequence-specific treatment resulted in distinct products; by design, the treatment would yield a 17 nt DNA product (visualized) and a 6 nt product (too short for PAGE visualization). To further demonstrate the DSN-mediated target recycling, we performed additional validation experiments, as detailed in our response to comment 4.

Fig. R9. Comparison of DNA cleavage mechanisms in RNA-DNA heteroduplexes by DSN and BstNI. Schematic illustration of mechanism differences (left). DSN recognizes the RNA-DNA heteroduplex and cleaves the DNA strand in a random, repeated manner to yield short DNA products (≤ 6 nt). BstNI, as an example of enzymes that mediate sequence-specific cleavage, recognizes specific cut site (CC/WGG) in the RNA-DNA heteroduplex to cleave the DNA into distinct products. Experimental validation (right). We incubated identical RNA-DNA heteroduplexes with DSN or BstNI and characterized the DNA products through polyacrylamide gel electrophoresis (10% PAGE). DSN yielded no distinct DNA product while BstNI produced a distinct band that corresponded to the specific BstNI cleavage sequence (17 nt DNA product). Note that DNA products ≤ 6 nt are too short for the PAGE visualization.

4. Regarding the previous question, in Supplementary Figure 1c, the authors should provide evidence that the increase in signal over time is not due to the slow rate of reaction.

This is another important point. To evaluate if the observed signal increase over time is due to DSN-mediated target recycling (and not due to the slow rate of reaction), we performed additional validation experiments. Specifically, we performed a two-step sequential reaction (**Fig. R10**, and **new Supplementary Fig. 5b**). In the first step, we prepared the reaction with a high concentration of target RNA, mixed with the DNA probes in a 1:1 ratio to form RNA-DNA heteroduplexes, and added in excess DSN to initiate signaling. The target RNA and DSN concentrations were kept consistent to conditions as used in **Supplementary Fig. 1c**. Notably, this configuration has a high rate of reaction but does not support target recycling (due to the 1:1 formed RNA-DNA heteroduplexes). By measuring the resultant cumulative signal generated over time, we observed

that the signal saturated very quickly (<3 mins) and attained only <10% of the original absolute signal as measured in **Supplementary Fig. 1c** (indicated by the red box). To evaluate if additional signal could be generated through target recycling, without introducing additional target RNA, we performed sequential reactions by topping up with different reagents: 1) excess specific DNA probes to enable target cycling, 2) excess scrambled DNA probes, or 3) buffer control. All additions were matched in volume. Only the addition of specific DNA probes resumed signal generation and reached a comparably high signal, while the scrambled control and buffer control generated minimal signal increase over time. These results thus confirm that the observed signal increase is primarily due to RNA target recycling, as the the signal output is determined by the amount of DNA probes available to form heteroduplexes with a fixed input of RNA target.

Fig. R10. Two-step sequential reaction. In the first step, RNA-DNA heteroduplexes (made from mixing RNA targets and DNA probes in a 1:1 ratio) were reacted with excess DSN. The resultant cumulative signal was measured over time. In the second step, after signal saturation, we topped up the reaction with excess DNA probes, scrambled DNA probes, or buffer. Only the addition of specific DNA probes resumed signal generation and reached a comparably high signal (as indicated by the red box) while the scrambled control and buffer control generated minimal signal increase. All measurements were performed in triplicate, and the data are displayed as mean \pm s.d.

5. A workflow illustration/schematics for the clinical sample would be appreciated.

We thank the reviewer for the helpful suggestion. We have included a workflow for the clinical sample (**Fig. R11**, also **new Supplementary Fig. 14**).

Fig. R11. Clinical sample workflow. Clinical samples were first collected and processed for plasma extraction. Plasma samples were lysed prior to EZ-READ measurement.

6. Is absolute quantification of RNA levels in human samples possible from the digital counting, perhaps using a calibration curve?

To address this query, we spiked known amounts of *Arabidopsis thaliana ath-miR159a*, a target absent in human, into human plasma. When measured with the EZ-READ transducer against *ath-miR159a*, we determined the number of fluorescent wells and showed that the RNA targets can be quantified through a linear calibration curve (Fig. R12, also new Supplementary Fig. 9c).

Fig. R12. EZ-READ calibration curve. Human plasma was spiked with varying amounts of *Arabidopsis thaliana ath-miR159a*, a target absent in human samples. The spiked plasma was incubated with transducers against *ath-miR159a*, and the number of fluorescent wells was determined to establish a calibration curve for target quantitation.

7. The authors should include a comparison of the sensitivity of their technique to similar systems for microRNA detection based on FRET.

We thank the reviewer for the helpful suggestion. We have included a comparison table to evaluate the EZ-READ platform with FRET-based miRNA detection systems (Table R1, also new Supplementary Table 4).

Table R1. Comparison with FRET-based systems for miRNA detection.

	EZ-READ	FRET without amplification	FRET with amplification	
		Hybridization (Molecular beacon) ¹⁻³	Isothermal amplification (RT-LAMP) ^{4,5}	RT-qPCR (TaqMan/SYBR Green) ⁶⁻⁸
Detection mechanism	Direct detection of RNA with linear signal amplification based on regenerative transduction of HRP@ZIF-8 nanoparticles	Direct detection of RNA without amplification based on fluorescence restoration of internally quenched fluorophore	Isothermal exponential amplification of cDNA with detection of fluorescence signal via hydrolysis probes or intercalating dyes	Thermal cycling-based exponential amplification of cDNA with detection of fluorescence signal via hydrolysis probes or intercalating dyes
Sample processing requirement	Low	Moderate (RNA extraction)	High (RNA extraction, reverse transcription)	High (RNA extraction, reverse transcription)
Amplification condition	Isothermal	Isothermal	Isothermal	Thermal cycling
Compatibility for short RNA	High (direct detection)	High (direct detection)	Low (extensive processing to incorporate additional sequence)	Low (extensive processing to incorporate additional sequence)
Sequence design stringency	Low (single sequence)	Low (single sequence)	High (sets of four or six primer sequences)	Moderate (sets of two or three primer and probe sequences)
Detection limit (copies)	~9	3000 – 3 x 10 ⁸	6 x 10 ⁵ – 8.4 x 10 ⁸	~7 – 1.2 x 10 ⁶
Time taken	~30 min	~1 h	1 – 2 h	2 – 4 h
Equipment requirement	Moderate	Moderate	Moderate	High

¹ Caputo, T.M., Battista, E., Netti, P.A. and Causa, F. *ACS Appl. Mater. Interfaces*, 11(19), 17147-17156 (2019)

² Hwang, J.Y. et al. *ACS Sens.*, 3(12), 2651-2659 (2018)

³ Hu, J. et al. *Biomaterials*, 183, 20-29 (2018)

⁴ Li, C., Li, Z., Jia, H. and Yan, J. *Chem. Commun.*, 47(9), 2595-2597 (2011)

⁵ Williams, M.R., Stedtfeld, R.D., Stedtfeld, T.M., Tiedje, J.M. and Hashsham, S.A. *Biomed. Microdevices*, 19, 1-8 (2017)

⁶ Chen, C. et al. *Nucleic Acids Res.*, 33(20), e179-e179 (2005)

⁷ Jung, S. et al. *Biosens. Bioelectron.*, 163, 112301 (2020)

⁸ Ge, Q. et al. *Anal. Methods*, 6(22), 9101-9107 (2014)

8. A paragraph discussing the outlook of this study, including the required steps to obtain regulatory approval for clinical testing and the challenges associated with commercializing this method, should be included in the discussion.

This is another excellent suggestion. We have updated the discussion accordingly. The EZ-READ is a direct and versatile platform for multiplexed profiling of diverse RNA markers. This is especially appealing for minimally-invasive characterization of inaccessible diseases (e.g., GBM) and could pave new clinical opportunities for patient stratification and personalized treatment. To enable technology translation, regulatory clearance and commercialization strategy are critical for its maturation and adoption. For regulatory approval, the platform could either be deployed as a laboratory-developed test or an in vitro diagnostic (IVD) test and this decision has extensive regulatory implications [Natalia, A., Zhang, L., Sundah, N.R., Zhang, Y. and Shao, H. *Nat. Rev. Bioeng.* (2023)]. Laboratory-developed tests can only be used in dedicated Clinically Laboratory Improvement Amendment (CLIA)-certified laboratories and are thus exempted from extensive regulatory oversight. In comparison, IVD tests require much more stringent regulatory clearance (e.g., analytical validation, clinical validation and clinical utility) for distributed use as standalone assays. To facilitate commercialization, multifaceted factors need to be considered upfront and progressively. These include clinical use cases, scale-up manufacturing, quality management as well as reimbursement model, amongst others, and require extensive collaborations across multiple stakeholders (e.g., scientists, clinicians, commercial developers and policymakers).

9. The microfluidic chamber used in this study is impressive. It is a new design, and the authors should emphasize its advantages more clearly in the main text.

We thank the reviewer for the helpful suggestion. We have updated the text to further empathize the advantages of the microfluidics. Specifically, the microfluidic platform features a fractal branching network of microchannels leading to individual reaction wells. This design enables the following properties. First, through repeat division and subdivision of channel branches, the microchannels leading to individual reaction wells have the same length and fluidic resistance; this ensures even reagent loading (e.g., HRP@ZIF-8 particles) into the wells. Second, the fractal branching design is scalable. By increasing the order of branching, the microfluidics can be readily expanded to increase the number of microwells for diverse applications. Third, by self-priming with negative pressure (vacuum pre-packaging), various reagents can be sequentially loaded and compartmentalized in the microwells, without requiring sophisticated liquid handling systems. In fact, the introduction of air flow can effectively seal off individual microwells to enable independent reactions and digital counting.

10. A short video of sample loading and video from microscope of the sample fluorescence change over time would be a nice addition.

We thank the reviewer for the helpful suggestion. We have included short videos on sample loading and fluorescence change over time (**new Supplementary Movies 1–2**).

Other minor comments:

1. The authors should consider analyzing their use of the word "faithful" in the manuscript. "Reliable" may be a more appropriate term, as faithful suggests loyalty and is likely unsuitable for describing non-living objects or methods.

We have corrected the text accordingly.

2. The negative aspects of PCR are overstated in the introduction. For example, things like "exquisite sequence design" are not issues, as they only have to be done once for a particular RNA.

We have corrected the text accordingly.

REVIEWERS' COMMENTS

Reviewer #1 (Remarks to the Author):

The authors have addressed questions re: IDH mutation status in their cohort and the performance of the EZREAD strategy. Future work should increase the number of patients in the study since only 3 patients with IDH mutant status were studied. If you tried to do the TERT assay and were not able to get it to work, that should be stated.

Reviewer #2 (Remarks to the Author):

The authors have addressed the raised comments, which is suitable for the publication in this journal.

Reviewer #3 (Remarks to the Author):

The authors have put great effort into addressing all my concerns. The questions on target recycling and calibration curves were very convincingly addressed with new experiments. They also have put a lot of effort into clarifying the cleavage mechanism, comparison with other systems, and clinical workflow with detailed figures, tables, and movie files. I would like the authors to add the information sample characterization of GBM (grade 4) provided in the rebuttal, to the main text in the "Clinical analysis of GBM diagnosis and subtyping" section. Beyond this particular point, I find no further concerns or comments to raise, and the revised paper appears to be solid.

REVIEWER COMMENTS

We thank the reviewers for their advice which have strengthened our manuscript.

Reviewer #1 (Remarks to the Author):

The authors have addressed questions re: IDH mutation status in their cohort and the performance of the EZREAD strategy. Future work should increase the number of patients in the study since only 3 patients with IDH mutant status were studied. If you tried to do the TERT assay and were not able to get it to work, that should be stated.

We thank the reviewer for the valuable feedback. In our last revision, we focused on IDH mutation status as it is a major bifurcation in the WHO2021 classification of glioma tumors. We did not perform the TERT assay due to a lack of relevant clinical information in our current patient population. We have included an updated discussion in the current manuscript on future developments with respect to TERT mutation profiling and evaluation in larger patient cohorts.

Reviewer #2 (Remarks to the Author):

The authors have addressed the raised comments, which is suitable for the publication in this journal.

We thank the reviewer for the positive comment.

Reviewer #3 (Remarks to the Author):

The authors have put great effort into addressing all my concerns. The questions on target recycling and calibration curves were very convincingly addressed with new experiments. They also have put a lot of effort into clarifying the cleavage mechanism, comparison with other systems, and clinical workflow with detailed figures, tables, and movie files. I would like the authors to add the information sample characterization of GBM (grade 4) provided in the rebuttal, to the main text in the "Clinical analysis of GBM diagnosis and subtyping" section. Beyond this particular point, I find no further concerns or comments to raise, and the revised paper appears to be solid.

We thank the reviewer for the positive comment and have included the information on sample characterization of GBM (grade 4) in the revised text.